# Differential Geometric Aspects of Parametric Estimation Theory for States on Finite-Dimensional *C*^∗^-Algebras

**DOI:** 10.3390/e22111332

**Published:** 2020-11-23

**Authors:** Florio M. Ciaglia, Jürgen Jost, Lorenz Schwachhöfer

**Affiliations:** 1Max Planck Institute for Mathematics in the Sciences, 04103 Leipzig, Germany; jjost@mis.mpg.de; 2Faculty of Mathematics, TU Dortmund University, 44221 Dortmund, Germany; lschwach@math.tu-dortund.de

**Keywords:** information geometry, estimation theory, Fisher–Rao metric tensor, Bures–Helstrom metric tensor, Cramer–Rao bound, Helstrom bound, Symmetric Logarithmic Derivative, Differential Geometry of *C*^∗^-algebras

## Abstract

A geometrical formulation of estimation theory for finite-dimensional $C^{*}$-algebras is presented. This formulation allows to deal with the classical and quantum case in a single, unifying mathematical framework. The derivation of the Cramer–Rao and Helstrom bounds for parametric statistical models with discrete and finite outcome spaces is presented.

## 1. Introduction

The purpose of this work is to present the formulation of estimation theory in the framework of $C^{*}$-algebras, with particular attention to differential geometric aspects. Although estimation theory is a well-developed subject both in the classical case of probability distributions [1,2,3,4] and the quantum case of density operators [5,6,7,8,9,10,11,12], and even if quantum estimation theory builds upon classical estimation theory, there is no unifying picture for these subjects. By unifying picture, we mean a mathematical framework in which estimation theory is placed in such a way that the classical and quantum cases appear as particular instances of the general theory. We believe that such a formulation may be helpful in obtaining a better understanding of the similarities and differences of classical and quantum estimation theory. This idea may be considered as the driving force of this work.

Roughly speaking, the main problem estimation theory tries to address is to infer the value of some parameters characterizing the state of the “physical system” under investigation by the theoretical manipulation of the outcomes of experiments performed on such system.

In the classical case, the state of the system is described by a probability distribution on the space of outcomes of the experiment, and the goal of estimation theory is to infer the value of some parameters characterizing the true probability distribution describing the system (e.g., the mean and/or variance for the case of Gaussian distributions) from the outcomes of the experiment. As such, estimation theory is well developed both in its asymptotic and non-asymptotic regimes. Arguably, one little black spot of the theory is that the parameter spaces characterizing the probability distributions under study are usually taken to be homeomorphic to open subsets of some finite-dimensional Euclidean space. Even if this assumption is justified in most of the models, it necessarily introduces some simplifications related with the “nice” structure of the parameter spaces as smooth manifolds. As an example, the existence of global coordinates often lead to the definition of objects that are coordinate-dependent (see, for instance, in ([3] ch. 4), where it is clearly stated that the notion of unbiased estimator developed there is coordinate-dependent). We believe it is healthy to formulate the theory in order to avoid these issues and better comprehend the coordinate-independent aspects of the theory. This geometric attitude already proved itself useful in classical Newtonian, Lagrangian, and Hamiltonian mechanics [13,14,15,16,17,18], in thermodynamics and statistical physics [19,20,21,22], and in quantum mechanics [23,24,25,26]. Clearly, there already have been efforts to formulate classical estimation theory in this direction [27,28,29,30], and here we try to encapsulate the spirit of these works in our formulation of estimation theory on $C^{*}$-algebras.

On the other hand, in the quantum case, the state of the system is no longer a probability distribution, it is a density operator on the Hilbert space associated with the system. This adds, at the same time, complexity and richness to the problem of estimation. A first layer of added complexity refers to the need of a statistical interpretation of a given quantum state. Since the dawn of quantum mechanics, the issue of the physical interpretation of Schrödinger’s wavefunction was recognized to be a fundamental question. The idea of interpreting the square modulus of the wavefunction as a probability distribution paved the way to the statistical interpretation of quantum states through what is now called the Born rule [31]. Essentially, the Born rule describes a “procedure” to associate a probability distribution on a suitable outcome space with a given quantum state. Clearly, this depends on both the quantum state and the choice of the outcome space, and this means that there is more than one way to associated probability distributions with quantum states. From the mathematical point of view, the choice of the statistical interpretation is described by a positive operator-valued measure (POVM) on the Hilbert space of the system [10,32]. Accordingly, in order to set up the estimation problem for a given parametric model of quantum states, we need to operate a preliminary choice concerning the POVM “inducing” the statistical interpretation. Of course, this choice influences the estimation problem we set up, and different choices in general lead to different solutions of the associated estimation problem. All this obviously adds a layer of complexity to the estimation problem, but, simultaneously, it opens new possibilities to outperform classical limits of estimation because of the peculiar features of quantum states (e.g., entanglement). Indeed, in the quantum case it is possible to give a precise mathematical meaning to the assertion “measuring one copy N times is less informative than measuring N copies one single time” [12,33,34,35,36]. This assertion relies on the phenomenon of entanglement which is absent in the classical realm, and thus highlights an important difference between the classical and quantum estimation theory.

As mentioned before, the goal of this article is to introduce a theoretical framework that allows us to treat the classical and quantum case simultaneously. Specifically, our choice is to consider the theory of $C^{*}$-algebras as the backbone of our construction because both probability distributions and quantum states may be realized as linear functionals on suitable $C^{*}$-algebras. In the case of probability distributions, this is basically the duality between probability measures and functions given by the Riesz theorem. For quantum states, this comes directly from the axiomatic structure of the theory. The main difference between the two cases is that the algebras involved are commutative in the former case and non-commutative in the latter. In this general framework, probability distributions and quantum states represent different realizations of the notion of ***state*** on a $C^{*}$-algebra A. The space of states S is a convex subset of the dual space of A, and the study of its differential geometry is a fascinating subject. The rich algebraic structure of $C^{*}$-algebras translates into a rich geometrical structure for their spaces of states that is perfectly suited for the formulation of parametric estimation theory.

The use of $C^{*}$-algebras as a theoretical framework to study the geometry of quantum states is not new [37,38,39,40,41,42,43,44,45,46,47,48,49,50,51,52,53,54,55,56]. However, the focus was essentially always on the algebra of bounded linear operators on the Hilbert space of the quantum system, and not on a generic $C^{*}$-algebra. While this restriction may seem not particularly relevant for most practical purposes, it is certainly so from the theoretical point of view. Indeed, some recent developments [57,58,59,60,61,62] point out the possibility of describing quantum systems whose associated $C^{*}$-algebras are groupoid algebras, and thus are in principle more general than the algebra of bounded linear operators. Consequently, a reformulation of the well-known results for an arbitrary $C^{*}$-algebra appears to be useful.

On the other hand, in the classical case, the explicit use of $C^{*}$-algebras to investigate the geometry of probability distributions is essentially absent. To the best of our knowledge, the only (very nice) exceptions are the works in [63,64,65]. However, the point of view of these works is different from ours because they consider probability distributions as particular elements of a $C^{*}$-algebras, while we consider them as particular linear functionals.

Another reason why we believe it would be useful to consider the framework of $C^{*}$-algebras is that the space of states of a $C^{*}$-algebra is an example of space of states of ***general probabilistic theories*** [66,67]. Therefore, the study of the differential geometry of the space of states of $C^{*}$-algebras, and in particular the study of parametric estimation theory in this context, represents a first step toward the study of these subjects in the more comprehensive framework of general probabilistic theories. This intermediate step may be useful because states on $C^{*}$-algebras benefit from the rich algebraic structure of the algebras they act upon, while states in general probabilistic theories do not necessarily have such a rich algebraic background to rely on. Consequently, a first study of the richer case may lead to results that can be later generalized to the less rich case once an appropriate and judicious process of extrapolation is pursued.

We confine ourselves to the case of finite-dimensional $C^{*}$-algebras because, at this preliminary stage, we want to avoid the technical difficulties with which the infinite-dimensional case is filled. Indeed, we are now interested in exposing the basic aspects of the theory in order to have a solid background on which future works can rely on. In the infinite-dimensional case, the technical difficulties would often obscure the conceptual aspects, and this unavoidably leads to be less communicative. Moreover, it is even not yet clear what are the geometrical players on the fields when infinite dimensions are considered because there is no general consensus on which are the most appropriate manifolds of states to consider in this case (see the works in [4,23,68,69,70,71,72,73,74,75,76,77,78] for some examples).

Incidentally, the restriction to the finite-dimensional case seems to affect more the classical case, rather than the quantum case. Indeed, classical estimation theory essentially deals with parametric models of probability distributions on spaces which are neither discrete nor finite (think for instance to normal distributions), and these cases are naturally associated with infinite-dimensional $C^{*}$-algebras. The case of parametric models of probability distributions on discrete and finite spaces is usually less studied because it seldom presents itself in applications. In the context of quantum information theory, the situation is quite the opposite, and the vast majority of the models considered refer to quantum system with a finite-dimensional Hilbert space, and thus with an associated finite-dimensional $C^{*}$-algebra. The infinite-dimensional case usually deals only with pure-state models for which the underlying manifold of states is rather friendly, being the Hilbert manifold of a complex projective space associated with a separable, complex Hilbert space.

The content of this work builds on well-known and established results in the context of both classical and quantum estimation theory. However, the presentation of these results in the unifying framework of $C^{*}$-algebras is essentially new, as are the proofs of some results. We believe that this attitude may be particularly useful in future research dealing with the infinite-dimensional case and in dealing with the comparison of classical and quantum methods. Accordingly, this work should be considered more as a first, preliminary step in a research program aimed at the understanding of the unification of classical and quantum estimation theory rather than an exposition of a finite theory, and the focus of the work is more on the discussion of general structures rather than on the presentation of specific examples.

The article is structured as follows. In Section 2, some differential geometric aspects of finite-dimensional $C^{*}$-algebras and of their spaces of states are recalled. In Section 3, the notion of parametric model of states on a $C^{*}$-algebra A is introduced and the notion of Symmetric Logarithmic Derivative used in quantum information theory is generalized to the $C^{*}$-algebraic setting. In Section 4, the notion of parametric statistical model associated with a given parametric model of states is introduced. This notion represents the bridge between the models of states on a possibly noncommutative $C^{*}$-algebra and the models of probability distributions used in classical estimation theory. Moreover, the notion of multiple round model and its geometrical properties are briefly discussed. In Section 5, the problem of estimation theory is formulated in the $C^{*}$-algebraic framework and the notion of manifold-valued estimator is recalled. In Section 6, a proof of the Cramer–Rao bound for manifold-valued estimators on finite outcome spaces is given following the work of Hendriks [29]. Finally, in Section 7, the generalization of the Helstrom bound used in quantum information theory to the $C^{*}$-algebraic framework is given.

## 2. Differential Geometric Aspects of the Space of States

We start with a brief summary of $C^{*}$-algebras [79,80,81,82]. Let *A* be a complex algebra with identity I. If there is an anti-linear map †:A→A such that ${\left(a^{†}\right)}^{†}=a$ for all a∈A, and such that ${\left(\mathrm{ab}\right)}^{†}=b^{†}a^{†}$ for all a,b∈A, then † is called an ***involution*** and (A,†) an ***involutive algebra***. If there is a norm ∥·∥ on *A* turning it into a Banach space satisfying the additional relations ∥ab∥≤∥a∥∥b∥ and $∥aa^{†}{∥\;\leq \;∥a∥}^{2}$ for all a,b∈A, then (A,†,∥·∥) is called a $C^{*}$***-algebra***, and, for the sake of notational simplicity, it will be denoted simply by A.

An element a∈A is called ***self-adjoint*** if $a=a^{†}$. The space of self-adjoint elements in A is denoted as $A_{sa}$. It is a real Banach space whose dual space is denoted as V, and there is a direct sum decomposition
(1)$$A=A_{sa}⊕ı\;A_{sa},$$
where *ı* is the imaginary unit.

An element b∈A is called ***positive*** if there exists a∈A such that $b=a^{†}a$. Clearly, a positive element b is self-adjoint, and it can be proved that there is a unique self-adjoint element s such that $b=s^{2}$.

An element g∈A is called ***invertible*** if there is another element written as $g^{-1}$ such that $g\;g^{-1}=g^{-1}\;g=I$. The set of invertible elements in A is denoted as G, and it is a real Banach–Lie group; the Banach–Lie algebra of which is A endowed with the commutator [83,84]. An element u∈G is called ***unitary*** if $u^{-1}=u^{†}$. The set of unitary elements in A is denoted as U and it is a real Banach–Lie subgroup of G, called the *unitary group of A*, whose Banach–Lie algebra is the subspace $ı\;A_{sa}$ in the decomposition (Equation 1) endowed with the commutator inherited from A [83,84].

Let $A^{*}$ be the complex Banach dual of A. An element $\xi \in A^{*}$ is called a ***self-adjoint*** linear functional if $\xi \left(a^{†}\right)=\bar{\xi (a)}$. The set of self-adjoint linear functionals is precisely the real Banach dual V of $A_{sa}$. A self-adjoint linear functional ω is called ***positive*** if ω(a)≥0 for every positive element a∈A. A positive element ω is called ***faithful*** if ω(a)=0 implies a=0 for all positive elements in A. The set of positive elements is denoted as P. A positive linear functional ρ is called a ***state*** if ρ(I)=1. The set of states is denoted as S.

In the following, we will focus only on finite-dimensional $C^{*}$-algebras. Given a self-adjoint element $a\in A_{sa}$, we write $f_{a}$ for the linear function on V given by
(2)$$f_{a}\left(\xi \right)\;=\;\xi \left(a\right)\;,$$
as well as for its restrictions to the various submanifolds of V we will introduce below (with an evident abuse of notation).

There is a group action of G on S given by [41,68]
(3)$$\rho_{g}\;:\;\;\rho_{g}\left(c\right)\;=\;\frac{\rho (g^{†}\;c\;g)}{\rho (g^{†}\;g)}\;\equiv \;\Phi \left(g,\rho \right)\;\;\;\;\forall \;c\in A,$$
and the space of states S decomposes into the disjoint union of orbits of the G-action, and evidently, each such orbit is a homogeneous space.

Recalling that A endowed with the commutator is the Lie algebra of G, the fundamental vector fields of Φ are labelled by elements of A. Recalling (Equation 1), we write an element in A as a+ıb where $a,b\in A_{sa}$, and *ı* is the imaginary unit. Accordingly, we write $\Gamma_{\mathrm{ab}}$ for the fundamental vector field associated with $\frac{1}{2}\left(a+ıb\right)$. A direct computation shows that the tangent vector $\Gamma_{\mathrm{ab}}\left(\rho \right)$, identified with a self-adjoint linear functional in V because the orbit O is an immersed submanifold of V, is given by
(4)$$\left(\Gamma_{\mathrm{ab}}\;f_{c}\right)\left(\rho \right)\;=\;\left(\Gamma_{\mathrm{ab}}\left(\rho \right)\right)\left(c\right)\;=\;\rho \left(\left\{a,c\right\}\right)-\rho \left(a\right)\rho \left(c\right)+\rho \left(\left[\left[b,c\right]\right]\right)\;\;\;\;\forall \;c\in A,$$
where {,} and [[,]] denote, respectively, the Jordan product and the Lie product in A given by
(5)$$\begin{array}{cc} \left\{c,d\right\} & \;:=\;\frac{1}{2}\;\left(\mathrm{cd}+\mathrm{dc}\right)\\ \left[[c,d]\right] & \;:=\;\frac{1}{2ı}\;\left(\mathrm{cd}-\mathrm{dc}\right). \end{array}$$Note that {,} and [[,]] preserve $A_{sa}$, and actually turn it into a Jordan–Lie algebra [85,86].

We set
(6)$$\begin{array}{cc} Y_{a} & \;:=\;\Gamma_{a0}\\ X_{b} & \;:=\;\Gamma_{0b}, \end{array}$$
and we call $Y_{a}$ a gradient vector field (the origin of the name will be explained below) and $X_{b}$ a Hamiltonian vector field. It is not hard to show that the Hamiltonian vector fields give an anti-representation of the Lie algebra of the group U⊂G of unitary element of G [41,68]. This Lie algebra anti-representation integrates to a left action of U on O given by the restriction of Φ to U.

If we fix a basis ${\left\{e^{j}\right\}}_{j=1,...,N}$ of self-adjoint elements in A (where dim(A)=N), we may introduce the structure constants $d_{l}^{jk}$ and $c_{l}^{jk}$ of the Jordan and Lie products in Equation (Equation 5) by setting
(7)$$\begin{array}{cc} \left\{e^{j},e^{k}\right\} & \;=\;d_{l}^{jk}\;e^{l}\\ {}[[e^{j},e^{k}]] & \;:=\;c_{l}^{jk}\;e^{l}. \end{array}$$Then, the gradient and Hamiltonian vector fields are easily seen to be given by
(8)$$\begin{array}{cc} Y_{a} & \;=\;\left(d_{l}^{jk}a_{k}x^{l}-f_{a}\;x^{j}\right)\;\frac{\partial }{\partial x^{j}}\\ X_{b} & \;:=\;c_{l}^{jk}b_{k}x^{l}\;\frac{\partial }{\partial x^{j}}, \end{array}$$
where ${\left\{x^{j}\right\}}_{j=1,...,N}$ is the Cartesian coordinate system on V associated with the dual basis ${\left\{e_{j}\right\}}_{j=1,...,N}$ of ${\left\{e^{j}\right\}}_{j=1,...,N}$.

**Example** **1**(The probability simplex). *If we endow $C(X_{n})$ with the involution given by complex conjugation, and with the supremum norm, it is not hard to prove that it is a $C^{*}$-algebra. We denote this $C^{*}$-algebra as $C_{n}$. Let $e^{j}\in C_{n}$ be the “delta function” at $x_{j}\in X_{n}$ (i.e., $e^{j}\left(x_{k}\right)=\delta_{k}^{j}$), then ${\left\{e^{j}\right\}}_{j=1,..,n}$ is clearly a basis for $C_{n}$ (seen as a vector space) made up of positive, self-adjoint elements, and we have*
(9)$$\sum_{j=1}^{n}\;e^{j}\;=\;1_{n},$$*where $1_{n}$ is the identity element in $C_{n}$ (i.e., the identity function on $X_{n}$). Consequently, we can build the dual basis ${\left\{e_{j}\right\}}_{j=1,..,n}$, and a state ρ on $C_{n}$ is easily seen to be written as*
(10)$$\rho \;=\;p^{j}\;e_{j},$$*where the real numbers $p^{j}=\rho \left(e^{j}\right)$ are non-negative and are subject to the constraint*
(11)$$\sum_{j=1}^{n}\;p^{j}\;=\;1.$$*From this, we conclude that the space of states S of $C_{n}$ may be identified with the n-simplex $\Delta_{n}$. In the following, whenever we deal with $C_{n}$, we will identify a state ρ on $C_{n}$ with a probability distribution in $\Delta_{n}$ and write p instead of ρ.*

*Let $I_{k}\subseteq X_{n}$ be a subset with k≤n elements, and let ρ be a state on $C_{n}$ such that $p^{j}\neq 0$ if and only if $x_{j}\in I_{k}$. Then, it is not hard to check that the orbit O of the group G of invertible elements in $C_{n}$ (see Equation (Equation 3)) through ρ coincides with the set of all those states $\varrho =q^{j}e_{j}$ such that $q^{j}\neq 0$ if and only if $x_{j}\in I_{k}$. In particular, the open interior $\Delta_{n}^{+}$ of the n-simplex may be identified with the orbit of G through the state p with $p^{j}=\frac{1}{n}$ for all j=1,...,n.*

*As $C_{n}$ is Abelian, it is not hard to see that the action of the unitary group U⊂G is trivial, and thus the Hamiltonian vector fields vanish identically. On the other hand, a direct computation shows that the structure constants $d_{l}^{jk}$ of the Jordan product with respect to the basis ${\left\{e^{j}\right\}}_{j=1,..,n}$ vanish unless j=k=l, in which case they are *1*.*


**Example** **2**(The space of density matrices). *Consider the complex algebra $M_{n}:=M_{n}\left(C\right)$ of complex-valued, (n×n) matrices. There is an involution *†* on $M_{n}$ given by the composition of transposition with component-wise complex conjugation. By exploiting the trace operation, it is possible to define a norm on $M_{n}$ given by ${∥a∥}^{2}\;=\mathrm{Tr}\left(a^{†}a\right)$, and we obtain a $C^{*}$-algebra which will be denoted by $M_{n}$. Moreover, it is easily seen that $M_{n}$ is isomorphic to the algebra B(H) of bounded linear operators on an n-dimensional complex Hilbert space H. This isomorphism depends on the choice of an orthonormal basis in H, but, in the context of quantum information theory, this is in general not very limiting because a preferred choice of basis, called **computational basis** [87], is often tied to the physics of the problem under investigation.*
*As $M_{n}$ is finite-dimensional, it is isomorphic with its dual space, and an isomorphism is provided by the trace operation. Specifically, a linear functional ξ on $M_{n}$ is identified with an element $\hat{\xi }\in M_{n}$ by means of*
(12)$$\xi \left(a\right)\;:=\;\mathrm{Tr}\left(\hat{\xi }\;a\right).$$
*Then, it follows that a state ρ on $M_{n}$ may be identified with a positive semi-definite matrix $\hat{\rho }\in M_{n}$ with unit trace. Any such matrix is usually referred to as a density matrix.*

*It is not hard to prove that the orbits of G are classified by the rank of the associated density matrices [38,40,45,68,88]. Specifically, every orbit O is made up of states the associated density matrices of which have fixed rank. In particular, we have the orbit of states whose density matrices have unit rank which is the orbit of pure states (the extremal points of the convex space of states) which is diffeomorphic to the complex projective space ${\mathrm{CP}}^{n}$, and the orbit of states whose density matrices have full rank (invertible) which is the orbit of faithful states. Note that the latter is an open subset of the affine space of self-adjoint linear functionals giving 1 when evaluated on the identity $I_{n}$ of $M_{n}$.*

*If we introduce a basis ${\left\{\sigma^{j}\right\}}_{j=0,...,n^{2}-1}$ on $M_{n}$ in such a way that $\sigma^{0}$ coincides with the identity element $I_{n}\in M_{n}$, and that $\sigma^{j}$ is self-adjoint and satisfies $\mathrm{Tr}(\sigma^{j})=0$ for all j≠0, we can build its dual basis ${\left\{\sigma_{j}\right\}}_{j=0,...,n^{2}-1}$, and it follows that a state ρ may be written as*
(13)$$\rho \;=\;\frac{1}{n}\left(\sigma_{0}+x^{j}\;\sigma_{j}\right),$$
*where $j=1,...,n^{2}-1$ and $x^{j}\in R$. Clearly, the fact that ρ must be a state imposes some constraints on the values of $x^{j}$ depending on the fact that $\rho (a^{†}a)$ must be non-negative. There is no general closed formula to express these constraints for arbitrary n>2.*

*For the case n=2 (also known as the **qubit**), it is customary to select $\sigma^{1},\sigma^{2},\sigma^{3}$ to be the so-called Pauli matrices*
(14)$$\sigma^{1}=\left(\begin{array}{cc} 0 & 1\\ 1 & 0 \end{array}\right)\;\sigma^{2}=\left(\begin{array}{cc} 0 & -ı\\ ı & 0 \end{array}\right)\;\sigma^{3}=\left(\begin{array}{cc} 1 & 0\\ 0 & -1 \end{array}\right),$$
*where ı is the imaginary unit. Then, ρ is a state if and only if*
(15)$$\delta_{jk}x^{j}x^{k}\;\leq \;1\;.$$
*This identifies a ball in the three-dimensional space spanned by the Pauli matrices which is known as the **Bloch ball**. In this case, there are only two orbits of the group G of invertible elements in $M_{n}$, namely, the density matrices lying on the surface sphere (the **pure states**) and the density matrices in the interior of the ball (the **faithful states**).*


According to the work in [41], the gradient vector fields provide an overcomplete basis of the tangent space at each point in every orbit O. Furthermore, on every O we may define a Riemannian metric tensor G given by
(16)$$G_{\rho }\left(Y_{a}\left(\rho \right),Y_{b}\left(\rho \right)\right)\;=\;\rho \left(\left\{a,b\right\}\right)-\rho \left(a\right)\rho \left(b\right)\;,$$
and $Y_{a}$ is the gradient vector field associated with the smooth function $f_{a}$ (see Equation (Equation 2)) by means of G. This metric tensor is invariant with respect to the action of the unitary group in the sense that
(17)$$\Phi_{U}^{*}G\;=\;G\;\forall \;U\in U,$$
where $\Phi_{U}$ is the diffeomorphism given by
(18)$$\Phi_{U}\left(\rho \right)\;:=\;\Phi \left(U,\rho \right)\;.$$However, G is not invariant under the action of all of G.

The metric tensor G turns out to be the $C^{*}$-algebraic version of some well-known and relevant metric tensors when explicit cases are considered [41]. For instance, if $A=C_{n}$ and $O=\Delta_{n}^{+}$, then G coincides with the Fisher–Rao metric tensor. If A=B(H) and O≅CP(H) is the orbit of pure states), then G coincides with the Fubini–Study metric tensor. If A=B(H) and O is the orbit of faithful states, then G coincides with the Bures–Helstrom metric tensor.

According to the work in [41], the geodesic of G starting at ρ∈O with initial tangent vector $v\in T_{\rho }O$ reads
(19)$$\nu_{\rho }^{v}\left(t\right)\;=\;\cos^{2}\left(|v|t\right)\;\rho +\frac{\sin^{2}\left(|v|t\right)}{{\left|v\right|}^{2}}\;\rho_{v}+\frac{\sin(2|v|t)}{2|v|}\;\rho_{\left\{v\right\}},$$
where
(20)$$\begin{array}{cc} v\;=\;Y_{a}\left(\rho \right)\;\mathrm{for}\;\mathrm{some} & \;a\in A_{sa}\;|\;\;\rho \left(a\right)\;=\;0,\\ {\left|v\right|}^{2}\;=\;G_{\rho }\left(v,v\right) & \;=\;\rho \left(a^{2}\right),\\ \rho_{v}\left(b\right)\;:=\;\rho \left(a\;b\;a\right)\; & \forall \;\;a\in A_{sa},\\ \rho_{\left\{v\right\}}\left(b\right)\;:=\;\rho \left(\{a,\;b\}\right)\; & \forall \;\;a\in A_{sa}. \end{array}$$The geodesic $\nu_{\rho }^{v}\left(t\right)$ remains inside the space of states S for all t∈R, but it also exits and enters the orbit O containing the initial state ρ at multiple times [41].

## 3. Parametric Models of States on $C^{*}$-algebras

Motivated by the classical theory of parametric estimation, we will now introduce the notion of a parametric model of states on a finite-dimensional $C^{*}$-algebra, and then reformulate the theory of parametric estimation in this theoretical framework. This will allow for the simultaneous handling of the classical and the quantum case.

**Definition** **1.***A **parametric model** of states on a (finite-dimensional) $C^{*}$-algebra A is a triplet (M,j,O) where M is a smooth manifold, O⊂S is a G-orbit in S (see Section 2), and j:M→O is a smooth map. If j is injective, we say that the model is **identifiable***.

Some comments are in order. First of all, we fix the codomain of j to be an orbit of states O because, as will be clear below, we want to exploit the differential geometric aspects of O itself. In practice, a vast part of the models considered in the literature falls in this category. For instance, in quantum information geometry, it is customary to deal with parametric models consisting only of pure states, or only of invertible density operators. In principle, it would also be possible to consider a more general case in which j is a smooth map of *M* into the Banach space V of self-adjoint linear functionals in such a way that j(M)⊂S, and *M* intersects different orbits of states. This line of thought would require a different way to handle geometrical properties of the space of states in relation with the parameter manifold, based, for example, on the methodology introduced in [4,71] for the classical case. This line of reasoning may be useful in the transition to the infinite-dimensional case where the smooth structure of the orbits O is in general not guaranteed, and we plan to address this and related questions in the future.

Concerning the identifiability of a model, it may seem at first glance reasonable to consider only identifiable models, but we will show that there are well-known and “simple” parametric models of quantum states (e.g., qubit models) for which either this assumption is not satisfied, or it leads to difficulties with the statistical interpretation of the model.

Now, we turn our attention to the geometrical objects that *M* inherits by means of the smooth map j. Indeed, once we have the smooth map j, a symmetric, covariant (0,2) tensor is naturally obtained on *M* by considering the Riemannian metric G on O introduced before and taking its pullback
(21)$$G^{M}\;:=j^{*}G$$
to *M* with respect to j. This gives a tensor on *M* which “feels” the possible non-commutativity of A and gives the “correct” tensor in the classical case.

Indeed, if A is Abelian, then O is diffeomorphic to the open interior of a suitable simplex, G is the Fisher–Rao metric tensor [41], and $G^{M}$ is the pullback of the Fisher–Rao metric tensor to the manifold *M* seen as a model of probability distributions [3].

On the other hand, if A is the algebra B(H) of bounded linear operators on a finite-dimensional, complex Hilbert space H and O is the manifold of pure states, then O is diffeomorphic to the complex projective space CP(H) associated with H, G is the Fubini–Study metric [41] on O=CP(H), and $G^{M}$ is the quantum counterpart of the Fisher–Rao metric tensor on the manifold *M* seen as a model of pure quantum states [89]. Moreover, if O is the manifold of faithful states, then G is the Bures–Helstrom metric tensor [41], and $G^{M}$ may be read as a quantum counterpart of the Fisher–Rao metric tensor on the manifold *M* seen as a model of faithful quantum states [90].

We will now introduce the $C^{*}$-algebraic version of the Symmetric Logarithmic Derivative (SLD) introduced in quantum estimation theory by Helstrom in [6]. For this purpose, note that every tangent vector at ρ∈O may be expressed in terms of gradient vector fields, that is, given ρ∈O, for every tangent vector $V_{\rho }\in T_{\rho }O$ there exists a self-adjoint element $a\in A_{sa}$ depending on $V_{\rho }$ such that
(22)$$V_{\rho }\;=\;Y_{a}\left(\rho \right)\;.$$Consequently, if we consider a tangent vector $v_{m}\in T_{m}M$, it makes sense to ask for the gradient vector field $Y_{a}$ on O such that
(23)$$T_{m}j\left(v_{m}\right)\;=\;Y_{a}\left(\rho_{m}\right)\;,$$
where $\rho_{m}:=\rho \left(j\left(m\right)\right)$. The gradient vector field $Y_{a}$ in general depends on both the point m∈M and the tangent vector $v_{m}$. The tangent vector $Y_{a}\left(\rho_{m}\right)$ satisfying Equation (Equation 23) is called the SLD of $v_{m}$ at $\rho_{m}$.

To appreciate the link with the standard definition of the SLD, let us consider a parametric model (M,j,O) where A=B(H), O is the manifold of faithful states (invertible density operators), *M* is an open submanifold of R, and j is a suitable smooth map. Setting $v_{m}=\partial_{t}\left(m\right)$ where $\partial_{t}$ is the restriction to *M* of the vector field generating the group structure of R, a direct computation shows that the solution of Equation (Equation 23) coincides with the Symmetric Logarithmic Derivative (SLD) of the work in [6]. Indeed, $\partial_{t}$ is the infinitesimal generator of $m_{t}=m+t$, and considering an arbitrary function $f_{b}$ on O, we have
(24)$$\langle df_{b}\left(\rho_{m}\right),\;T_{m}j\left(v_{m}\right)\rangle \;=\;\frac{d}{dt}\;{\left(\mathrm{Tr}\left({\hat{\rho }}_{m_{t}}\;b\right)\right)}_{t=0}\;=\;\mathrm{Tr}\left(\frac{d}{dt}\;{\left({\hat{\rho }}_{m_{t}}\right)}_{t=0}\;b\right)\;\;\;\forall \;b\in A_{sa}$$
so that Equation (Equation 23) may be alternatively written as
(25)$$\frac{d}{dt}\;{\left({\hat{\rho }}_{m_{t}}\right)}_{t=0}\;=\;\left\{{\hat{\rho }}_{m},\;a_{m}\right\}\;=\;\frac{1}{2}\left({\hat{\rho }}_{m}\;a_{m}+a_{m}\;{\hat{\rho }}_{m}\right),$$
where
(26)$$a_{m}\;=\;a-{\hat{\rho }}_{m}\left(a\right)\;I,$$
which is precisely the definition of the SLD (see also Equation (Equation 3) in [90] and Equations (3.4) and (3.14) in [91] for the multiparametric case). This justifies the interpretation of Equation (Equation 23) as the $C^{*}$-algebraic generalization of the SLD embracing also the multiparametric quantum and classical cases.

**Example** **3**(A pure state qubit model). *Consider the algebra $M_{2}$ of the qubit (see Example 2). Take the one-parameter group of unitary elements generated by the element $ı\sigma_{3}$ according to*
(27)$$u_{\gamma }\;=\;e^{\frac{ı}{2}\gamma \sigma^{3}},$$*where γ∈R. Then, consider the orbit $O≅{\mathrm{CP}}^{2}$ of pure states on $M_{2}$, set M=R, and consider the map $j_{R}:M\to O$ given by*
(28)$$\rho_{\gamma }\;\equiv \;j_{R}\left(\gamma \right)\;:=\;\Phi \left(u_{\gamma },\rho \right),$$*where *Φ* is the action of G⊃U given in Equation (Equation 3), and*
(29)$$\rho \;=\;\frac{1}{2}\left(\sigma_{0}+\sigma_{1}\right).$$*A direct computation shows that*
(30)$$\rho_{\gamma }\;=\;\frac{1}{2}\left(\sigma_{0}+\cos\left(\gamma \right)\;\sigma_{1}-\sin\left(\gamma \right)\;\sigma_{2}\right)$$*and that $j_{R}$ is smooth. Clearly, $j_{R}$ is not injective, and thus the parametric model $\left(R,j_{R},{\mathrm{CP}}^{2}\right)$ is not identifiable. However, the parametrization given in Equation (Equation 30) is useful in quantum estimation theory when an experimental realization of the parametric model is constructed in terms of a spin interacting with a magnetic field. In this case, γ=tB where t is the time parameter of the dynamical evolution and B is the strenght of the magnetic field. Then, the fact that the model is not identifiable depends on the dynamical evolution being periodic.*
*Now, let us consider the vector field V on M=R generating translations. This vector field is complete, and provide a basis of the tangent space $T_{\gamma }M$ at each γ∈M. Moreover, V is the infinitesimal generator of the action of the Abelian Lie group G=R on M=R given by*
(31)ψ(ζ,γ):=γ+ζ∀ζ∈G,γ∈M.
*The group G acts also on ${\mathrm{CP}}^{2}$ by means of*
(32)$$\Psi \left(\zeta ,\rho \right)\;:=\;\Phi \left(U_{\zeta },\rho \right),$$
*where *Φ* is the action given in Equation (Equation 3). The fact that *Ψ* is a group action follows from the fact that the map $\gamma \mapsto U_{\gamma }$ is a group homomorphism, that is, it satisfies*
(33)$$U_{\zeta_{1}}\;U_{\zeta_{2}}\;=\;U_{\zeta_{1}+\zeta_{2}}\;\;\forall \;\zeta_{1},\zeta_{2}\in G.$$
*The actions ψ and *Ψ* have a particular relation to one another, indeed, a direct computation shows that they are equivariant with respect to $j_{R}$, which means that*
(34)$$j_{R}\left(\psi (\zeta ,\gamma )\right)\;=\;\Psi \left(\zeta ,j_{R}\left(\gamma \right)\right)\;.$$
*This property is quite strong because it implies that the fundamental vector fields of the action of G on M=R are $j_{R}$-related with the fundamental vector fields of the action of G on ${\mathrm{CP}}^{2}$, which means that [92]*
(35)$$T_{\gamma }j_{R}\left(V_{\gamma }\right)\;=\;W_{\rho_{\gamma }}^{\zeta },$$
*where V is the fundamental vector field of ψ(ζ,γ) (i.e, the vector field generating the translation considered above), while W is the fundamental vector field of Ψ(ζ,ρ) (recall that, in this case, the exponential map from the Lie algebra of G to G itself is the identity). As*
(36)$$\Psi \left(\zeta ,\rho \right)\;=\;\Phi \left(U_{\zeta },\rho \right),$$
*the fundamental vector field W is easily seen to be the Hamiltonian vector field associated with $\sigma^{3}$ (see Equation (Equation 6)). This means that*
(37)$$\langle df_{b},\;T_{\gamma }j_{R}\left(V_{\gamma }\right)\rangle \;=\;\langle df_{b},\;W_{\rho_{\gamma }}\rangle \;=\;\rho_{\gamma }\left([\left[\sigma^{3},\;b\right]]\right)\;.$$
*Consequently, regarding the SLD, Equation (Equation 23) leads us to look for the self-adjoint element a satisfying*
(38)$$\rho_{\gamma }\left(\{a,\;b\}\right)-\rho_{\gamma }\left(a\right)\;\rho_{\gamma }\left(b\right)\;=\;\rho_{\gamma }\left([\left[\sigma^{3},\;b\right]]\right)\;$$
*for all self-adjoint elements $b\in M_{2}$. Passing from $\rho_{\gamma }$ to its density matrix ${\hat{\rho }}_{\gamma }$, we see that Equation (Equation 38) is equivalent to*
(39)$$\left\{{\hat{\rho }}_{\gamma },\;a\right\}-\mathrm{Tr}\left({\hat{\rho }}_{\gamma }a\right)\;\hat{\rho }\;=\;\left[\left[{\hat{\rho }}_{\gamma },\;\sigma^{3}\right]\right].$$
*We write*
(40)$$a\;=\;a_{0}\sigma^{0}+a_{1}\sigma^{1}+a_{2}\sigma^{2}+a_{3}\sigma^{3},$$
*where $a_{j}\in R$ for all j=0,1,2,3. A direct computation exploiting the properties of the Pauli matrices shows that $a_{0}$ is arbitrary (as it should be because of the very definition of gradient vector field), $a_{3}=0$, while $a_{1}$ and $a_{2}$ must satisfy*
(41)$$a_{1}\sin\left(\gamma \right)+a_{2}\cos\left(\gamma \right)\;=\;-1.$$
*Clearly, this means that a and thus the SLD are not uniquely defined.*

*Concerning the covariant tensor $G^{R}$, we have*
(42)$$G^{R}\;=\;j_{R}^{*}G$$
*by definition. As $j_{R}$ is an immersion and G is a Riemannian metric, then $G^{R}$ is a Riemannian metric (i.e., it is positive and invertible). Moreover, setting*
(43)$$\begin{array}{cc} \psi_{\zeta }\left(\gamma \right) & \;:=\;\psi (\zeta ,\gamma )\\ \Psi_{\zeta }\left(\rho \right) & \;:=\;\Psi \left(\zeta ,\rho \right)\;=\;\Phi \left(U_{\zeta },\rho \right), \end{array}$$
*we immediately obtain*
(44)$$\psi_{\zeta }^{*}G^{R}\;=\;\psi_{\zeta }^{*}j_{R}^{*}G\;=\;{\left(j_{R}\circ \psi_{\zeta }\right)}^{*}G\;=\;{\left(\Psi_{\zeta }\circ j_{R}\right)}^{*}G\;=\;j_{R}^{*}\;\Psi_{\zeta }^{*}G\;=\;j_{R}^{*}\;\Phi_{U_{\zeta }}^{*}G\;=\;j_{R}G\;=\;G^{R}$$
*where we used Equation (Equation 34) in the fourth equality, and Equation (Equation 17) in the sixth equality. Therefore, we conclude that $G^{R}$ is invariant with respect to the action of the Lie group G=R on M=R given by translation, and thus must be proportional to the Euclidean metric tensor.*


**Example** **4**(A mixed state qubit model). *Consider the algebra $M_{2}$ of the qubit (see Example 2). Consider the orbit O of faithful states, set $M=R^{+}\times R^{+}$, and define the map $j_{M}$ as*
(45)$$\rho_{\gamma ,\zeta }\;\equiv \;j_{M}\left(\gamma ,\zeta \right)\;:=\;\frac{1}{2}\;\left(\sigma_{0}+e^{-\zeta \;\gamma }\left(\cos\left(\gamma \right)\;\sigma_{1}-\sin\left(\gamma \right)\;\sigma_{2}\right)\right).$$*A direct computation shows that this map is smooth. Quite interestingly, the parametric model $\left(M,j_{M},O\right)$ has a physical origin which is connected with the dynamics of open quantum systems. The dynamics of such systems is governed by the so-called Gorini–Kossakowski–Lindblad–Sudarshan (GKLS) equation [93,94,95,96,97,98]. In particular, choosing the infinitesimal generator L of this linear equation to be the dephasing channel, the dynamical evolution evolution generated by L is such that the initial (pure) state ρ given in Equation (Equation 29) evolves according to the right-hand-side of Equation (Equation 45), where $\frac{\gamma }{2}$ plays the role of the time parameter while 2ζ is the dephasing parameter ([40] ex. 2). Note that the initial pure state is evolved into a mixed (faithful) state as soon as the time parameter is greater than 0.*
*This model has been recently considered in the context of quantum parameter estimation in the presence of nuisance parameters [99].*

*Let us now consider the vector fields V and W on M generating the local one-parameter groups of local diffeomorphisms*
(46)$$\begin{array}{cc} \phi_{t}\left(\gamma ,\zeta \right) & \;=\;(\gamma +t,\zeta )\\ \psi_{t}\left(\gamma ,\zeta \right) & \;=\;(\gamma ,\zeta +t). \end{array}$$
*Clearly, these vector fields are not complete on M; however, they provide a basis of tangent vectors at each point of M. A direct computation shows that*
(47)$$\begin{array}{cc} \langle df_{b},\;T_{\gamma ,\zeta }j_{M}\left(V_{\gamma ,\zeta }\right)\rangle & \;=\;-e^{-\zeta \gamma }\left(\left(\sin(\gamma )+\zeta \cos(\gamma )\right)\;b_{1}+\left(\cos(\gamma )-\zeta \sin(\gamma )\right)\;b_{2}\right)\\ \langle df_{b},\;T_{\gamma ,\zeta }j_{M}\left(W_{\gamma ,\zeta }\right)\rangle \; & \;=\;-\gamma e^{-\zeta \gamma }\left(\cos\left(\gamma \right)\;b_{1}-\sin\left(\gamma \right)\;b_{2}\right), \end{array}$$
*from which we conclude that $j_{M}$ is an immersion. Then, Equation (Equation 23) implies that the SLD $Y_{a^{V}}\left(\rho_{\gamma ,\zeta }\right)$ and $Y_{a^{W}}\left(\rho_{\gamma ,\zeta }\right)$ of V and W at (γ,ζ), respectively, are found as the solutions of*
(48)$$\begin{array}{cc} \langle df_{b},\;T_{\gamma ,\zeta }j_{M}\left(V_{\gamma ,\zeta }\right)\rangle & =\langle df_{b},\;Y_{a^{V}}\left(\rho_{\gamma ,\zeta }\right)\rangle =\rho_{\gamma ,\zeta }\left(\{a^{V},b\}\right)-\rho_{\gamma ,\zeta }\left(a^{V}\right)\;\rho_{\gamma ,\zeta }\left(b\right)\\ \langle df_{b},\;T_{\gamma ,\zeta }j_{M}\left(E_{\gamma ,\zeta }\right)\rangle & =\langle df_{b},\;Y_{a^{W}}\left(\rho_{\gamma ,\zeta }\right)\rangle =\rho_{\gamma ,\zeta }\left(\{a^{W},b\}\right)-\rho_{\gamma ,\zeta }\left(a^{W}\right)\;\rho_{\gamma ,\zeta }\left(b\right) \end{array}$$
*for all self-adjoint elements $b\in M_{2}$. A direct computation leads to*
(49)$$\begin{array}{cc} a^{V} & \;=\;a_{0}^{V}\sigma^{0}-\left(e^{-\zeta \gamma }\sin\left(\gamma \right)+\frac{\zeta \cos(\gamma )}{2\sinh(\zeta \gamma )}\right)\sigma^{1}+\left(\frac{\zeta \sin(\gamma )}{2\sinh(\zeta \gamma )}-e^{-\zeta \gamma }\cos\left(\gamma \right)\right)\sigma^{2}\\ a^{W} & \;=\;a_{0}^{W}\sigma^{0}-\frac{\gamma }{2\sinh(\zeta \gamma )}\left(\cos\left(\gamma \right)\sigma^{1}-\sin\left(\gamma \right)\sigma^{2}\right). \end{array}$$
*Note that, apart from the coefficients $a_{0}^{V}$ and $a_{0}^{W}$, which are arbitrary because they do not affect the expression of the associated gradient vector field, the SLD associated with V and W are uniquely defined at each point of M. This is due to the fact that the model is a model of faithful states. Moreover, note that $[a^{V},a^{W}]\neq 0$, and thus there is no unital, Abelian $C^{*}$-subalgebra of $M_{2}$ that contains both $a^{V}$ and $a^{W}$. This will have an impact on the attainability of the Helstrom bound.*

*As $G^{M}=j_{M}^{*}G$, we immediately obtain (see Equation (Equation 16))*
(50)$$G_{\gamma ,\zeta }^{M}\left(V_{\gamma ,\zeta },V_{\gamma \zeta }\right)=G_{\rho_{\gamma ,\zeta }}\left(Y_{a^{V}}\left(\rho_{\gamma ,\zeta }\right),Y_{a^{V}}\left(\rho_{\gamma ,\zeta }\right)\right)=\rho_{\gamma ,\zeta }\left(\{a^{V},a^{V}\}\right)-{\left(\rho_{\gamma ,\zeta }\left(a^{V}\right)\right)}^{2},$$
*and similarly for $G_{\gamma ,\zeta }^{M}\left(V_{\gamma ,\zeta },W_{\gamma \zeta }\right)$ and $G_{\gamma ,\zeta }^{M}\left(W_{\gamma ,\zeta },W_{\gamma \zeta }\right)$. Then, as V and W provide a basis of tangent vectors at each point in M, the tensor $G^{M}$ can be computed to be*
(51)$$G^{M}\;=\;\left(e^{-2\zeta \gamma }+\frac{\zeta^{2}}{e^{2\zeta \gamma }-1}\right)d\gamma ⊗d\gamma +\left(\frac{\zeta \gamma }{e^{2\zeta \gamma }-1}\right)d\gamma {⊗}_{S}d\zeta +\left(\frac{\gamma^{2}}{e^{2\zeta \gamma }-1}\right)d\zeta ⊗d\zeta .$$


**Example** **5**(Lie group and Lie algebra parametric models). *Motivated by the model in Example and by some of the models commonly used in the quantum context [5,100], we introduce the notion of a **Lie group parametric model** and of a **Lie algebra parametric model***.
*Let G be a Lie group which is realized as a Lie subgroup of the Lie group G of invertible elements in A, and let $\rho_{0}$ be a state in S. Set M=G and define the map $j_{G}:M\to O$, where O is the orbit containing $\rho_{0}$, by means of*
(52)$$j_{G}\left(g\right)\;:=\;\Phi \left(g,\rho_{0}\right).$$
*This map is clearly smooth and we call $\left(G,j_{G},O\right)$ a **Lie group parametric model**. If the fiducial state $\rho_{0}$ is such that*
(53)$$\Phi \left(g,\rho_{0}\right)\;=\;\rho_{0}\;\;⟺\;\;g=I\;\forall g\in G\;,$$
*then the model is identifiable.*

*As G is a subgroup of G, the left action of G on itself is related with the action of G on O determined by the restriction of *Φ* to G in the way expressed in Equation (Equation 34). Specifically, let ψ be the left action of G on itself. Define an action *Ψ* of G on O given by*
(54)Ψ(g,ρ):=Φ(g(g),ρ),
*where g∈G and g(g)∈G is the realization of g as an element of G. Then, g↦g(g) is a group homomorphism, that is, it satisfies*
(55)$$g\left(g_{1}g_{2}\right)\;=\;g\left(g_{1}\right)\;g\left(g_{2}\right),$$
*and thus it follows from Equations (Equation 52), (Equation 54), and (Equation 55) that*
(56)$$j_{G}\left(\psi (g,h)\right)\;=\;\Psi \left(g\left(g\right),j_{G}\left(h\right)\right),$$
*which means that the actions ψ and *Ψ* are equivariant with respect to $j_{G}$. This means that the fundamental vector fields of ψ are $j_{G}$-related with the fundamental vector fields of *Ψ* [92]. This instance may be helpful in computing the SLD adapting the steps outlined in Example 3.*

*If $\left(G,j_{G},O\right)$ is a Lie group parametric model and we consider another parameter manifold which is a smooth homogeneous space M=G/H of G admitting a global, smooth section η:M→G, then we can immediately build another parametric model $\left(M,j_{M},O\right)$ by setting $j_{M}:=j_{G}\circ \eta$. This may be helpful to obtain identifiable models. Indeed, if $\rho_{0}$ has a non-trivial isotropy group $G_{0}\subset G$, which is the set of all elements g∈G such that $\Phi \left(g,\rho_{0}\right)=\rho_{0}$, we have that $M=G/G_{0}$ is a smooth manifold. Then, if there is a smooth section η for M, the resulting parametric model will be identifiable. This is very similar to the notion of coherent state used in quantum theory [101,102].*

*Another relevant parametric models is obtained when we consider the Lie algebra g of G. In this case, we have the exponential map exp:g→G that can be exploited to define a parametric model. Specifically, let (G,j,O) be a Lie group parametric model. Then, defining $j_{g}:=j_{G}\circ \exp$, we immediately obtain the parametric model $\left(g,j_{g},O\right)$ which is referred to as a **Lie algebra parametric model**. If the Lie algebra g is commutative, then the exponential map is a group homomorphism when the Lie algebra is thought of as a group with respect to the vector sum, and we obtain an equivariance relation with respect to $j_{g}$ between the left action ψ of g on itself and its realization Ψ(v,ρ)=Φ(exp(v),ρ) as a group acting on O.*


## 4. Parametric Statistical Models of States on $C^{*}$-algebras

When an experiment is performed on a system in a given state ρ, we obtain an outcome lying in a given outcome space X which is associated with the measurement procedure. The state ρ is then “transformed” into a probability distribution on X in the sense that different repetitions of the same experimental procedure (i.e., preparation of the system in the state ρ followed by the measurement procedure with outcome space X) will produce in general different outcomes characterized by a probability distribution which is associated with the state ρ and with the measurement procedure adopted. In this work, we will always consider outcome spaces which are discrete and finite.

Given a discrete and finite outcome space $X_{n}$ with *n* elements, the statistical interpretation of the state ρ is encoded in a map $m^{*}:S⟶P\left(X_{n}\right)\equiv \Delta_{n}$, which we will assume to be convex in order to preserve one of the basic features of probabilities and states. From this, it follows that $m^{*}$ can be extended to a linear map $m^{*}:A^{*}⟶S\left(X_{n}\right)$, where S(X) is the vector space of signed measures on $X_{n}$. From the $C^{*}$-algebraic perspective, $S(X_{n})$ is the space of self-adjoint linear functionals on the Abelian $C^{*}$-algebra $C_{n}:=C\left(X_{n}\right)$ of complex-valued, continuous functions on $X_{n}$, and thus, as $m^{*}$ is continuous because $A^{*}$ and $S(X_{n})$ are finite-dimensional, we immediately obtain that there is a continuous linear map $m:C_{n}⟶A$ of which $m^{*}$ is the dual map. By construction, the map m must be such that its dual map $m^{*}$ sends the space of states of A into the space of states of $C_{n}$. One way to implement this condition is to require $m:C_{n}⟶A$ to be a unital, positive map between $C^{*}$-algebras, that is, a linear map preserving the identity and sending positive elements into positive elements (clearly, any such map sends self-adjoint elements into self-adjoint elements).

**Definition** **2.***A positive unital map $m:C_{n}\to A$ is defined to be a **measurement procedure***.

Specifically, given a finite and discrete outcome space $X_{n}$, we can always consider the basis of $C_{n}$ given by the elements ${\left\{e^{j}\right\}}_{j=1,...,n}$, where $e^{j}$ is the “delta function” at the *j*-th element of $X_{n}$. The measurement procedure m amounts to define the elements
(57)$$m^{j}\;:=\;m\left(e^{j}\right)\;\forall \;j=1,...,n,$$
in such a way that they satisfy
(58)$$\sum_{j=1}^{n}\;m^{j}\;=\;I,$$
and
(59)$$m^{j}\;\geq \;0\;\forall \;j=1,...,n.$$Essentially, we are considering a (discrete) POVM in the $C^{*}$-algebraic framework. The probability distribution $m^{*}\left(\rho \right)$ associated with the state ρ is characterized by the numbers
(60)$$p^{j}\;:=\;\left(m^{*}\left(\rho \right)\right)\left(e^{j}\right)\;=\;\rho \left(m(e^{j})\right)\;=\;\rho \left(m^{j}\right)\;.$$

Once a parametric model (M,j,O) is chosen, we immediately have the map
(61)$$j^{c}\;:=\;m^{*}\circ j:M\;⟶\;\Delta_{n}\;.$$We require this map to lie entirely in a given fixed orbit of states inside $\Delta_{n}$. Clearly, as every orbit in $\Delta_{n}$ is diffeomorphic to $\Delta_{k}^{+}$ for some k≠n (see Example 1), there is no loss of generality in requiring the codomain of $j^{c}$ to lie entirely inside the manifold $\Delta_{n}^{+}$ of faithful states on $C_{n}$. Indeed, if this is not the case, it suffices to redefine $X_{n}$ to be the subset $I_{k}$, exchange $C_{n}$ with $C(I_{k})$, and relabel *k* as *n*.

**Definition** **3.**
*Let (M,j,O) be a parametric model of states on a $C^{*}$-algebra A. A **measurement procedure**  m such that $j^{c}\left(M\right)\;:=\;m^{*}\circ j\left(M\right)\subseteq \Delta_{n}^{+}$ is called **regular** for (M,j,O).*


Once a regular measurement procedure m for (M,j,O) is chosen, we are ready to build a parametric statistical model (in the sense of information geometry [2,3,4]) associated with the parametric model (M,j,O).

**Definition** **4.**
*Let (M,j,O) be a parametric model of states on a $C^{*}$-algebra A, and let m be a regular measurement procedure for (M,j,O). Then, the triple $\left(M,j^{c},\Delta_{n}^{+}\right)$, with $j^{c}$ as in Equation (Equation 61), is defined to be the **parametric statistical model** associated with the parametric model (M,j,O) by means of the measurement procedure m.*


The open interior of the simplex $\Delta_{n}^{+}$ coincides with the space of faithful states on the finite-dimensional, commutative $C^{*}$-algebra $C_{n}$ of complex-valued continuous functions on the discrete n-point space Xn, and thus the Radon–Nikodym derivative of $p\in \Delta_{n}^{+}$ with respect to the counting measure on $X_{n}$ is well defined as a function on $X_{n}$, and it is called the probability density function of p. Clearly, being $j^{c}\left(M\right)\subseteq \Delta_{n}^{+}$, every element m∈M may be uniquely associated with the probability density function of $p_{m}=j^{c}\left(m\right)$. Moreover, for every $x\in X_{n}$, the function $p\left(m,x\right)=p_{m}\left(\left\{x\right\}\right)$ is a smooth function on *M* because $p_{m}$ is a linear functional on $C_{n}$ and $j^{c}$ is smooth, and its support does not depend on the chosen $x\in X_{n}$ because $j^{c}\left(M\right)\subseteq \Delta_{n}^{+}$. These regularity properties are particularly meaningful with respect to the Cramer–Rao bound discussed in Section 6.

**Remark** **1**(Classical statistical models). *In the specific case when the algebra A is commutative, i.e., $A=C_{n}$ for some n∈N, a **parametric model** (M,j,O) of states on $C_{n}$ is already a **parametric statistical model** by itself. Indeed, according to Example 1, the orbit O is diffeomorphic to the open interior $\Delta_{k}^{+}$ of a k-simplex with k≠n. Specifically, we have a subset $I_{k}\subseteq X_{n}$ of k elements, the $C^{*}$-algebra $C_{k}$ generated by the elements $e^{j}\in C_{n}$ with j such that $x_{j}\in I_{k}$, and O is diffeomorphic to the orbit of faithful states of $C_{k}$. Then, we have a “natural” measurement procedure $m:C_{k}\to C_{n}$ at our disposal given by the natural identification $i_{k}$ map of $C_{k}$ in $C_{n}$, and the map $j^{c}=m^{*}\circ j=i_{k}^{*}\circ j$ gives rise to the statistical model $\left(M,j^{c},\Delta_{k}^{+}\right)$ associated with (M,j,O). From this, it is clear that once we have the parametric model (M,j,O) we immediately have a “natural” parametric statistical model $\left(M,j^{c},\Delta_{k}^{+}\right)$ associated with it. No additional choices must be made.*

Exploiting the Riemannian geometry of $\Delta_{n}^{+}$, the parameter manifold *M* may be endowed with another symmetric, covariant (0,2) tensor which is in general different from the metric $G^{M}$ introduced before. Indeed, we may consider the Fisher–Rao Riemannian metric $G_{FR}$ on $\Delta_{n}^{+}$, which is the Riemannian metric tensor G associated with the Jordan product of the self-adjoint part of $C_{n}$ as described in Section 2, and then take its pullback
(62)$$G^{Mc}={\left(j^{c}\right)}^{*}G_{FR}$$
to *M* (the “c” stands for classical, or commutative). In this case, we obtain a symmetric covariant tensor on *M* which, unlike $G^{M}$ given by Equation (Equation 21), cannot feel the possible non-commutativity of A, and which is the pullback of the Fisher–Rao metric tensor on *M* thought of as a parametric statistical model in $\Delta_{n}^{+}$ along the lines of classical information geometry.

To accommodate multiple runs, say *N*, of the same experimental procedure on *N* identical and independent copies of the initial state, we introduce the parametric model $\left(M,j^{N},O^{N}\right)$, where $O^{N}$ is the manifold of states on the tensor product algebra
(63)$$A^{⊗N}:=A⊗\cdots ⊗A$$
containing the product states of the form $\rho_{1}⊗\cdots ⊗\rho_{N}$ with $\rho_{j}\in O$ for every j=1,...,N, and $j^{N}:M⟶O^{N}$ is given by
(64)$$j^{N}\left(m\right)\;:=\;j\left(m\right)\;⊗\;\cdots \;⊗\;j\left(m\right)\;\equiv \;\rho_{m}\;⊗\;\cdots \;⊗\;\rho_{m}\;\equiv \;\rho_{m}^{⊗N}.$$Clearly, we may endow *M* with the Riemannian metric $G^{MN}$ defined by
(65)$$G^{MN}\;:=\;{\left(j^{N}\right)}^{*}G^{N},$$
where $G^{N}$ denotes the canonical Riemannian metric on $O^{N}$ associated with the Jordan product on $A^{⊗N}$. As the smooth embedding $j^{N}$ has been defined in terms of a “multiplicative object”, namely, the tensor product, it is reasonable to expect that this multiplicative feature reflects also in the pullback metric. Indeed, below we will prove that
(66)$$G^{MN}\;=\;NG^{M}.$$

Performing *N* runs of an experiment provides us with a list of *N* outcomes, and we consider the outcome space
(67)$$X^{N}\;=\;X\;\times \;\cdots \;\times \;X.$$At this point, we must choose a measurement procedure $m^{N}:C_{n}^{⊗N}=C\left(X^{N}\right)⟶A^{⊗N}$ so that, setting $j^{cN}=m^{N}\circ j^{N}$, we can build a statistical model $\left(M,j^{cN},\Delta_{Nn}^{+}\right)$ in the obvious way. We may endow *M* with the Riemannian metric $G^{McN}$ defined by
(68)$$G^{McN}\;:=\;N{\left(j^{cN}\right)}^{*}G_{FR},$$
where $NG_{FR}$ is the Fisher–Rao metric tensor on $\Delta_{nN}^{+}$ (this either follows from standard arguments in classical information geometry, or by Proposition 1 below applied to the case where $A=C_{n}$).

**Proposition** **1.**
*With the notations introduced above, we have*
(69)$$G^{MN}\;=\;NG^{M}.$$


**Proof.** We start proving that, if $v_{m}\in T_{m}M$ is such that
(70)$$T_{m}j\left(v_{m}\right)\;=\;Y_{a}\left(\rho_{m}\right),$$
then it holds
(71)$$T_{m}j^{N}\left(v_{m}\right)\;=\;Y_{a^{N}}^{N}\left(\rho_{m}^{⊗N}\right),$$
where $Y_{a^{N}}^{N}$ is the gradient vector field on $O^{N}$ associated with
(72)$$a^{N}\;=\;a\;⊗\;I\;⊗\;\cdots \;⊗\;I+I\;⊗\;a\;⊗\;I\;⊗\;\cdots \;⊗\;I+\cdots +I\;⊗\;\cdots \;⊗\;I\;⊗\;a.$$Recall that simple elements of the form $b_{1}\;⊗\cdots \;⊗\;b_{N}$ generate $A^{⊗N}$, and thus, to prove Equation (Equation 71), it is sufficient to compute
(73)$$\langle df_{b_{1}⊗\cdots ⊗b_{N}}\left(\rho_{m}^{⊗N}\right),\;T_{m}j^{N}\left(v_{m}\right)\rangle \;=\;\langle d{\left(j^{N}\right)}^{*}f_{b_{1}⊗\cdots ⊗b_{N}}\left(m\right),\;v_{m}\rangle .$$Denoting by $m_{t}$ a smooth curve in *M* starting at *m* with initial tangent vector $v_{m}$, we have
(74)$$\begin{array}{cc} \langle d{\left(j^{N}\right)}^{*}f_{b_{1}⊗\cdots ⊗b_{N}}\left(m\right),\;v_{m}\rangle & \;=\;\frac{d}{dt}\;{\left(\rho_{m_{t}}^{⊗N}\left(b_{1}\;⊗\;\cdots \;⊗\;b_{N}\right)\right)}_{t=0}\;=\;\\ & \;=\;\frac{d}{dt}\;{\left(\rho_{m_{t}}\left(b_{1}\right)\;\cdots \;\rho_{m_{t}}\left(b_{N}\right)\right)}_{t=0}, \end{array}$$
from which Equation (Equation 71) follows applying the Leibniz rule and recalling that $T_{m}j\left(v_{m}\right)\;=\;Y_{a}\left(\rho_{m}\right)$.We now take $v_{m},w_{m}\in T_{m}M$ such that
(75)$$\begin{array}{cc} T_{m}j^{N}\left(v_{m}\right) & \;=\;Y_{a^{N}}^{N}\left(\rho_{m}^{⊗N}\right)\\ T_{m}j^{N}\left(w_{m}\right) & \;=\;Y_{b^{N}}^{N}\left(\rho_{m}^{⊗N}\right), \end{array}$$
with $a^{N}$ and $b^{N}$ as in Equation (Equation 72). Recalling that $G^{MN}={\left(j^{N}\right)}^{*}G^{N}$, and noting that
(76)$$G_{\rho_{m}^{⊗N}}^{N}\left(Y_{a^{N}}^{N}\left(\rho_{m}^{⊗N}\right),\;Y_{b^{N}}^{N}\left(\rho_{m}^{⊗N}\right)\right)\;=\;\rho_{m}^{⊗N}\left(\left\{a^{N},b^{N}\right\}\right)-\rho_{m}^{⊗N}\left(a^{N}\right)\;\rho_{m}^{⊗N}\left(b^{N}\right)$$
because of Equation (Equation 16), we have
(77)$$\begin{array}{cc} G_{m}^{MN}\left(v_{m},w_{m}\right) & \;=\;G_{\rho_{m}^{⊗N}}^{N}\left(Y_{a^{N}}^{N}\left(\rho_{m}^{⊗N}\right),\;Y_{b^{N}}^{N}\left(\rho_{m}^{⊗N}\right)\right)\\ & \;=\;\rho_{m}^{⊗N}\left(\{a^{N},\;b^{N}\}\right)-\rho_{m}^{⊗N}\left(a^{N}\right)\;\rho_{m}^{⊗N}\left(b^{N}\right)\;=\;\\ & \;=\;\left(N\rho_{m}\left(\left\{a,b\right\}\right)+N\left(N-1\right)\;\rho_{m}\left(a\right)\;\rho_{m}\left(b\right)\right)-N^{2}\;\rho_{m}\left(a\right)\;\rho_{m}\left(b\right)\;=\;\\ & \;=\;N\left(\rho_{m}\left(\left\{a,b\right\}\right)-\rho_{m}\left(a\right)\;\rho_{m}\left(b\right)\right)\;=\;\\ & \;=\;N\;G_{m}^{M}\left(v_{m},\;w_{m}\right)\; \end{array}$$
as desired. □

## 5. The Problem of Estimation Theory

The purpose of estimation theory is to manipulate the outcomes of experiments in such a way to obtain an estimate of the “true state” on which the experiment has been performed. This is done by means of a map $E:X_{n}⟶\;M$ called ***estimator***. In the following, we will always consider ***non-constant*** estimators.

Clearly, we need to come up with a way of establishing optimality for estimators. For this purpose, we introduce a smooth ***cost function***
C:M×M⟶R which is non-negative and vanishes only on the diagonal. The choice of the cost function is essentially left to the ingenuity of the theoretician, and it is difficult to outline a general selection methodology. However, in some cases, the choice of the cost function is suggested by the context.

Starting with a cost function *C*, and writing $E_{j}\equiv E\left(x_{j}\right)$ for the value of the estimator at the *j*-th element of the outcome space $X_{n}$, we introduce the function L:M×M⟶R given by
(78)$$L\left(m_{1},m_{2}\right)\;:=\;\sum_{j=1}^{n}\;C\left(m_{1},E_{j}\right)\;p^{j}\left(m_{2}\right)\;=\;\sum_{j=1}^{n}\;C\left(m_{1},E_{j}\right)\;\rho_{m_{2}}\left(m^{j}\right),$$
where $\left(p^{1}\left(m_{2}\right),\cdots ,p^{n}\left(m_{2}\right)\right)=j^{c}\left(m_{2}\right)\;=m^{*}\left(\rho_{m_{2}}\right)$, and m is the measurement procedure “generating” the statistical model $\left(M,j^{c},\Delta_{n}^{+}\right)$ associated with the parametric model (M,j,O) of states on A under investigation. It is clear from Equation (Equation 78) that if the cost function *C* is constant, then *L* does not actually depend on $m_{2}$, and the problem of estimation theory as will be now developed will lose meaning.

The function *L* may be seen as the expectation value of the real-valued, *M*-parametric random variable $C(m_{1},E\left(\cdot \right))$ on $X_{n}$ with respect to the *M*-parametric probability distribution $m(\rho_{m_{2}})$ on $X_{n}$. Therefore, *L* measures how centered is the probability distribution generated by $C(m_{1},E\left(\cdot \right))$.

Let $m_{*}\in M$ and denote by $L_{*}$ the function
(79)$$L_{*}\left(m\right)\;:=\;L\left(m,m_{*}\right).$$The estimator E is called ***stationary*** for the cost function *C* at $m_{*}$ if $L_{*}$ has an extremum at $m=m_{*}$, that is, if
(80)$$\left(VL_{*}\right)\left(m_{*}\right)\;=\;0$$
for all vector fields *V* on *M*. The estimator E is called ***unbiased*** for the cost function *C* at $m_{*}\in M$ if the function $L_{*}$ has a minimum at $m=m_{*}$, and it is called ***locally unbiased*** for the cost function *C* at $m_{*}$ if $L_{*}$ has a local minimum at $m=m_{*}$. In general, for a given cost function *C*, unbiased estimators need not exist.

Now, we may define an *M*-parametric self-adjoint element M in A setting
(81)$$M_{m_{1}}\;:=\;\sum_{j=1}^{n}\;C\left(m_{1},E_{j}\right)\;m^{j}\;.$$This element clearly depends also on the estimator E and on the measurement procedure m. Moreover, it allows us to write the function *L* as the expectation value of $M_{m_{1}}$ with respect to the state $\rho_{m_{2}}$ according to
(82)$$L\left(m_{1},m_{2}\right)\;=\;\rho_{m_{2}}\left(M_{m_{1}}\right).$$

The estimation problem may be approached from two different perspectives of increasing difficulty:the regular measurement procedure m is fixed, and the unknown of the problem is the estimator E, andboth the regular measurement procedure m and the estimator E are considered unknown.

Clearly, the first case reduces to the classical problem of estimation, and may be faced relying on well-known methods like the maximum likelihood estimator. The limit on the precision is then governed by the Cramer–Rao bound (see Section 6). The second case is definitely more difficult to address because the freedom in the choice of the regular measurement procedure adds another layer of complexity. However, in this case, the precision is governed by the Helstrom bound (see Section 7), and allows for a sharpening of the Cramer–Rao bound. Indeed, the freedom in choosing the measurement procedure reflects in the possibility of consider different “classical scenarios”, and choose the one with the lowest Cramer–Rao bound.

Unfortunately, for both forms of the problem, there is no algorithm to solve the problem in full generality, and a case-by-case analysis is mandatory.

**Remark** **2**(Stationary estimators for Euclidean cost function). *Suppose that M is explicitly realized as an n-dimensional submanifold of $R^{N}$ for some positive N∈N with n≤N. In this context, a common choice in parameter estimation theory is to consider the cost function C which is the Euclidean distance on $R^{N}\times R^{N}$ restricted to M×M. Specifically, we have*
(83)$$C\left(m_{1},m_{2}\right)\;:=\;\frac{1}{2}\;{\left|m_{1}-m_{2}\right|}^{2}\;,$$*so that the function L reads*
(84)$$L\left(m_{1},m_{2}\right)\;:=\;\frac{1}{2}\;\sum_{j=1}^{n}\;{\left|m_{1}-E_{j}\right|}^{2}\;p^{j}\left(m_{2}\right).$$*This type of cost function is called a **Euclidean cost function** for obvious reasons. Clearly, the Euclidean cost function C depends on the actual realization of the (a priori abstract) manifold M into a suitable $R^{N}$. In particular, because of Whitney’s embedding theorem, given a parameter manifold M we can always build a Euclidean cost function. Of course, the actual usefulness of such a cost function is in principle not clear and should be investigated case by case. However, it often happens in concrete models that the parameter manifold M is “naturally” immersed in some given $R^{N}$ by construction, and thus the Euclidean cost function unavoidably presents itself from the start.*

*If $\left\{\theta^{1},...,\theta^{n}\right\}$ is a local system of coordinates on M, it is easy to see that being stationary at $m_{*}$ is equivalent to (see Equation (Equation 80))*
(85)$$m_{*}^{k}\left(\theta \right)=E_{m_{*}\left(\theta \right)}\left[E^{k}\right]\;\forall k=1,...,N\;and\;\;\forall r=1,...,n,$$
*where $m_{1}^{k}$ is the smooth function on M obtained by composing the canonical immersion of M in $R^{N}$ with the canonical projection on the k-th factor, $E^{k}$ is the real-valued random variable on X obtained by composing E with the canonical immersion of M in $R^{N}$ and with the canonical projection on the k-th factor, and where $E_{m_{*}}\left[\cdot \right]$ denote the expectation value with respect to the M-parametric probability distribution $p_{m_{*}}$.*

*As C>0 for all $(m_{1},m_{2})\in M\times M$ unless $m_{1}=m_{2}$, in which case it vanishes, we see that a stationary estimator at m∈M is also locally unbiased at m∈M.*

*When M is an open subset of $R^{N}$ and $\left\{\theta^{1},...,\theta^{N}\right\}$ is a system of Cartesian coordinates, and when Equation (Equation 85) holds for all m∈M, we recover the standard definition of an unbiased estimator used in classical and quantum estimation theory ([3] ch. 4).*


## 6. The Cramer–Rao Bound

Here, we recall Hendrik’s derivation of the Cramer–Rao bound for estimators with values in a manifold [29] when the underlying outcome space is discrete and finite. This gives a clear geometric picture of the Cramer–Rao bound which does not depend on the existence of a privileged coordinatization of the parameter space *M* as it is the case in most of the existing literature (see, for instance, in ([3] ch. 4) where it is clearly stated that the notion of unbiased estimator developed there is coordinate-dependent, as well as in [28,29,30])

Let $\left(M,j,\Delta_{n}^{+}\right)$ be a parametric statistical model. We refer to Definition 4 and the paragraph right after it, as well as to Remark 1 for a discussion of the regularity properties satisfied by the model $\left(M,j,\Delta_{n}^{+}\right)$. Recall that the metric $G^{M}$ determined by Equation (Equation 21) coincides with the Fisher–Rao tensor on *M* as determined by standard methods of information geometry [1,2,3]. We assume that $G^{M}$ is invertible.

In order to obtain the generalized Cramer–Rao bound for a stationary estimator, we need to exploit the geometrical properties of the product structure of the manifold M×M. We will now recall these geometrical properties following ([103] sec. 2), to which we refer for the explicit proofs.

First of all, we note that there are two projections $\pi_{l}$ and $\pi_{r}$ from M×M to *M* given by
(86)$$\begin{array}{cc} \pi_{l}\left(m_{1},m_{2}\right) & \;:=\;m_{1}\\ \pi_{r}\left(m_{1},m_{2}\right) & \;:=\;m_{2}, \end{array}$$
and there is also the diagonal immersion $i_{d}$ of *M* into M×M given by
(87)$$i_{d}\left(m\right)\;:=\;\left(m,\;m\right)\;.$$Given a vector field *X* on *M*, we may define its left and right lift to be the vector fields $X_{l}$ and $X_{r}$ on M×M characterized by
(88)$$\begin{array}{cc} X_{l}\left(\pi_{l}^{*}f\right) & \;=\;\pi_{l}^{*}\left(X\left(f\right)\right)\\ X_{r}\left(\pi_{r}^{*}f\right) & \;=\;\pi_{r}^{*}\left(X\left(f\right)\right) \end{array}$$
for every smooth function *f* on *M*. It is possible to prove that every vector field *X* on *M* is $i_{d}$-related with the vector field $X_{l}+X_{r}$ on M×M ([103] sec. 2).

If E is a stationary estimator at $m_{*}\in M$ then $L_{*}$ has an extremum at $m_{*}$, and this is equivalent to
(89)$${\left(i_{d}\left(X_{l}L\right)\right)}_{m=m_{*}}\;=\;0$$
for all vector fields $X_{l}$ on M×M. We assume that E is a stationary estimator for all $m_{*}\in M$. This means that the function $L=i_{d}\left(X_{l}L\right)$ identically vanishes. Consequently, given an arbitrary vector field *Y* on *M*, we also have
(90)$$0\;=\;Y\left(i_{d}^{*}L\right)\;=\;Y\left(i_{d}^{*}\left(X_{l}L\right)\right)\;=\;i_{d}^{*}\left((Y_{l}X_{l}+Y_{r}X_{l})L\right),$$
which means
(91)$$i_{d}^{*}\left(Y_{l}X_{l}L\right)\;=\;-i_{d}^{*}\left(Y_{r}X_{l}L\right)\;.$$

As E is stationary at every $m_{*}$, it follows that the Hessian form $H_{*}$ of $L_{*}$ at $m_{*}$ is well defined and we have
(92)$$H_{*}\left(X\left(m_{*}\right),\;Y\left(m_{*}\right)\right)\;:=\;\left(Y\;X\;L_{*}\right)(m_{*}).$$A moment of reflection shows that
(93)$$\left(Y\;X\;L_{*}\right)\left(m_{*}\right)\;=\;\left(i_{d}^{*}\left(Y_{l}X_{l}L\right)\right)\left(m_{*}\right)$$
so that
(94)$$H_{*}\left(X\left(m_{*}\right),\;Y\left(m_{*}\right)\right)\;=\;-\left(i_{d}^{*}\left(Y_{r}X_{l}L\right)\right)\left(m_{*}\right)$$
because of Equation (Equation 91). Set
(95)$$\begin{array}{cc} C_{E_{j}}\left(m\right) & \;:=\;C(m,E_{j}) \end{array}$$
so that we have
(96)$$L\left(m_{1},m_{2}\right)\;:=\;\sum_{j=1}^{n}\;C_{E_{j}}\left(m_{1}\right)\;p^{j}\left(m_{2}\right)$$
and we obtain
(97)$$H_{*}\left(X\left(m_{*}\right),\;Y\left(m_{*}\right)\right)\;=\;-\sum_{j=1}^{n}\;\left(X_{l}C_{E_{j}}\right)\left(m_{*}\right)\;\left(Y_{r}p^{j}\right)\left(m_{*}\right)\;.$$Introducing the real-valued random variables on the probability space $\left(X_{n},p\left(m_{*}\right)\right)$ given by
(98)$$\begin{array}{cc} F_{X}^{*}\left(x_{j}\right) & \;:=\;\left(X_{l}C_{E_{j}}\right)\left(m_{*}\right)\\ G_{Y}^{*}\left(x_{j}\right) & \;:=\;\left(Y_{r}\ln\left(p^{j}\right)\right)\left(m_{*}\right), \end{array}$$
we can rewrite the right hand side of Equation (Equation 97) as
(99)$$H_{*}\left(X\left(m_{*}\right),\;Y\left(m_{*}\right)\right)\;=\;-E_{*}\left[F_{X}^{*}\;G_{Y}^{*}\right]\;,$$
where $E_{*}\left[\cdot \right]$ denotes the expectation value with respect to the probability measure $p(m_{*})$. The expression
(100)$${\langle F,G\rangle }_{*}\;:=\;E_{*}\left[F\;G\right]$$
is an inner product on the space of random variables on the probability space $(X_{n},p\left(m_{*}\right)$, and the Cauchy–Schwarz inequality may be applied to obtain
(101)$${\left(H_{*}\left(X\left(m_{*}\right),\;Y\left(m_{*}\right)\right)\right)}^{2}\;\leq \;E_{*}\left[F_{X}^{*}\;F_{X}^{*}\right]\;E_{*}\left[G_{Y}^{*}\;G_{Y}^{*}\right]\;.$$Then, a direct computation shows that
(102)$$\begin{array}{cc} E_{*}\left[G_{Y}^{*}\;G_{Y}^{*}\right] & \;=\;\sum_{j=1}^{n}\left(Y_{r}\ln\left(p^{j}\right)\right)\left(m_{*}\right)\left(Y_{r}\ln\left(p^{j}\right)\right)\left(m_{*}\right)\;p^{j}\left(m_{*}\right)\;=\;\\ & \;=\;G^{M}\left(Y\left(m_{*}\right),\;Y\left(m_{*}\right)\right). \end{array}$$

Next, we introduce the expression
(103)$$\begin{array}{cc} C(X\left(m_{*}\right),Y\left(m_{*}\right)) & \;:=\;E_{*}\left[F_{X}^{*}\;F_{Y}^{*}\right], \end{array}$$
which according to (Equation 98) implicitly contains the cost function *C*, so that we can write Equation (Equation 97) as
(104)$${\left(H_{*}\left(X\left(m_{*}\right),\;Y\left(m_{*}\right)\right)\right)}^{2}\;\leq \;C\left(X\left(m_{*}\right),X\left(m_{*}\right)\right)\;G^{M}\left(Y\left(m_{*}\right),\;Y\left(m_{*}\right)\right)\;.$$Clearly, C depends on the cost function *C* and the estimator E.

Now, fix $X_{m_{*}}\in T_{m_{*}}M$, and define the function $H:T_{m_{*}}M⟶R$ given by
(105)$$Y\left(m_{*}\right)\equiv Y_{m_{*}}\;\mapsto \;H\left(Y_{m_{*}}\right)\;:=\;H_{*}\left(X_{m_{*}},\;Y_{m_{*}}\right).$$This function admits a maximum on the unit sphere determined by the Fisher–Rao metric. Indeed, the Fisher–Rao unit sphere in $T_{m_{*}}M$ is compact because the Fisher–Rao metric is a Riemannian metric (positive). Let $Y_{m_{*}}^{0}$ be a point on which *H* is maximum. Then, we may always find a real number λ such that
(106)$$H\left(Y_{m_{*}}\right)\;=\;\lambda \;G^{M}\left(Y_{m_{*}}^{0},Y_{m_{*}}\right)\;,$$
so that
(107)$$H\left(Y_{m_{*}}^{0}\right)\;=\;\lambda \;G^{M}\left(Y_{m_{*}}^{0},Y_{m_{*}}^{0}\right)\;=\;\lambda$$
because $Y_{m_{*}}^{0}$ lies on the Fisher–Rao unit sphere.

With an evident abuse of notation, we denote by $H_{*}\left(X_{m_{*}}\right)$ the covector in $T_{m_{*}}^{*}M$ acting as
(108)$$\langle H_{*}\left(X_{m_{*}}\right),\;Z_{m_{*}}\rangle \;:=\;H_{*}\left(Z_{m_{*}},\;X_{m_{*}}\right)\;\;\;\forall \;Z_{m_{*}}\in \;T_{m_{*}}M,$$
and by $G^{M}\left(Y_{m_{*}}^{0}\right)$ the covector in $T_{m_{*}}^{*}M$ given by
(109)$$\langle G^{M}\left(Y_{m_{*}}^{0}\right),\;Z_{m_{*}}\rangle \;:=\;G^{M}\left(Y_{m_{*}}^{0},Z_{m_{*}}\right)\;\;\;\forall \;Z_{m_{*}}\in \;T_{m_{*}}M\;.$$Then, comparing Equation (Equation 105) with Equations (Equation 106), (Equation 108), and (Equation 109) allows us to conclude that
(110)$$H_{*}\left(X_{m_{*}}\right)\;=\;G^{M}\left(\lambda \;Y_{m_{*}}^{0}\right),$$
which, assuming $G^{M}$ to be invertible, is equivalent to
(111)$${\left(G^{M}\right)}^{-1}\left(H_{*}\left(X_{m_{*}}\right),\alpha_{m_{*}}\right)\;=\;\langle \alpha_{m_{*}},\;\lambda \;Y_{m_{*}}^{0}\rangle$$
for all covectors $\alpha_{m_{*}}\in T_{m_{*}}^{*}M$. In particular, setting $\alpha_{m_{*}}=H_{*}\left(X_{m_{*}}\right)$ we get
(112)$${\left(G^{M}\right)}^{-1}\left(H_{*}\left(X_{m_{*}}\right),H_{*}\left(X_{m_{*}}\right)\right)\;=\;\langle H_{*}\left(X_{m_{*}}\right),\;\lambda \;Y_{m_{*}}^{0}\rangle \;=\;\lambda \;H\left(Y_{m_{*}}^{0}\right)$$
because of Equations (Equation 108) and (Equation 105). Now, Equation (Equation 105) together with Equations (Equation 107) and (Equation 112) imply that
(113)$$\begin{array}{cc} {\left(H_{*}\left(Y_{m_{*}}^{0},\;X_{m_{*}}\right)\right)}^{2} & \;=\;{\left(H(Y_{m_{*}}^{0})\right)}^{2}\;=\;\lambda \;\;H\left(Y_{m_{*}}^{0}\right)\;=\;\\ & \;=\;{\left(G^{M}\right)}^{-1}\;\left(H_{m_{*}}\left(X_{m_{*}}\right),\;H_{m_{*}}\left(X_{m_{*}}\right)\right)\;. \end{array}$$Eventually, recalling that $Y_{m_{*}}^{0}$ lies on the Fisher–Rao unit sphere, Equations (Equation 104) and (Equation 113) lead us to the generalized Cramer–Rao bound
(114)$$C\left(X_{m_{*}},X_{m_{*}}\right)\;\geq \;{\left(G^{M}\right)}^{-1}\;\left(H_{*}\left(X_{m_{*}}\right),\;H_{*}\left(X_{m_{*}}\right)\right)\;.$$If the Hessian form of $L_{*}$ at $m_{*}$ is invertible, we define the covariance bivector Cov as
(115)$$\mathrm{Cov}\left(\xi_{m_{*}},\eta_{m_{*}}\right)\;:=\;C\left(H_{*}^{-1}\left(\xi_{m_{*}}\right),\;H_{*}^{-1}\left(\eta_{m_{*}}\right)\right),$$
where $\xi_{m_{*}},\eta_{m_{*}}\in T_{m_{*}}^{*}M$. We may then rewrite the generalized Cramer–Rao bound in terms of covectors. We proved the following.

**Proposition** **2.**
*Let $\left(M,j,\Delta_{n}^{+}\right)$ be a parametric statistical model for which $G^{M}$ is invertible. Let C be a cost function and let E be a stationary estimator for C at $m_{*}$. If the Hessian form of $L_{*}$ at $m_{*}$ is invertible, then we have the generalized Cramer–Rao bound*
(116)$$\mathrm{Cov}\left(\xi_{m_{*}},\xi_{m_{*}}\right)\geq \;{\left(G^{M}\right)}^{-1}\;\left(\xi_{m_{*}},\;\xi_{m_{*}}\right)\;$$
*for all $\xi_{m_{*}}\in T_{m*}^{*}M$.*


Let us stress that, because of the regularity properties satisfied by the model $\left(M,j,\Delta_{n}^{+}\right)$ (see Definition 4, the paragraph right after it, and Remark 1) and because of the assumed invertibility of $G^{M}$, the formulation of the Cramer–Rao bound given in Proposition 2 refers to the case in which the support of the considered probability distributions does not depend on the element *m* in the parameter manifold *M*. It is worth noting that in the literature, when the support of the considered probability density functions may depend on the parameter, there is still a version of the Cramer–Rao bound, the so-called Cramer–Rao–Leibniz bound, see, for instance, in [104].

A stationary estimator E which saturates the Cramer–Rao bound for every $v_{m}$ is called ***efficient***. The Cramer–Rao bound is related to the cost function *C* and to the estimator E; however, it is expressed in terms of the (inverse of the) Fisher–Rao metric tensor on *M* which is a geometrical object on *M*, which is completely independent of the cost function and the estimator. Note, however, that the expression (Equation 115) is invariant under rescaling the cost function *C*, because the expression C by (Equation 103) contains such a scaling factor quadratically, and this is canceled because the inverse of the Hessian enters quadratically into (Equation 115).

**Remark** **3**(The Cramer–Rao bound for Euclidean cost functions). *The “standard form” of the Cramer–Rao inequality used in classical information geometry is obtained when we M and the cost function C are as in Remark 2. In this case, a direct computation shows that, in local coordinates around $m_{*}$, the components Hessian form of $L_{*}$ at every stationary point are given by*
(117)$${\left(H_{*}\right)}_{jk}\;=\;\left(\delta_{rs}\;\frac{\partial m^{r}}{\partial \theta^{j}}\frac{\partial m^{s}}{\partial \theta^{k}}\right)\left(m_{*}\right).$$

*Assuming that M is open in the ambient manifold $R^{N}$, and taking $\left\{\theta^{1},...,\theta^{N}\right\}$ to be the Cartesian coordinates associated with the canonical projections of $R^{N}$ on R we immediately see that*
(118)$${\left(H_{*}\right)}_{jk}\;=\;\delta_{jk}.$$
*Therefore, writing*
(119)$${\left(\mathrm{Cov}(m_{*})\right)}^{jk}\;\equiv \;\mathrm{Cov}\left(d\theta^{j}\left(m_{*}\right),d\theta^{k}\left(m_{*}\right)\right),$$
*a direct computation shows that the covariance matrix ${\left(\mathrm{Cov}(m_{*})\right)}^{jk}$ at the point $m_{*}$ for which E is a stationary estimator reads*
(120)$$\begin{array}{cc} {\left(\mathrm{Cov}(m_{*})\right)}^{jk} & \;=\;E_{p_{*}}\left[\left(E^{j}-E_{p_{*}}\left[E^{j}\right]\right)\;\left(E^{k}-E_{p_{*}}\left[E^{k}\right]\right)\right] \end{array},$$
*which is essentially the form usually found in standard textbooks on estimation theory in statistics. The “standard form” of the Cramer–Rao bound follows immediately.*


## 7. The Helstrom Bound

The Cramer–Rao bound found in Section 6 applies to ***parametric statistical models***. As such, it depends only on the Fisher–Rao metric on *M*, which, in turn, depends on the properties of the Abelian algebra underlying the parametric statistical model. Accordingly, if $\left(M,j^{c},\Delta_{n}^{+}\right)$ is the parametric statistical model associated with a parametric model of states (M,j,O) on the possibly noncommutative $C^{*}$-algebra A, the Cramer–Rao bound for $\left(M,j^{c},\Delta_{n}^{+}\right)$ “does not feel” the possible noncommutativity of the algebra A. However, it is possible to formulate a bound which “feels” the possible non-commutativity of A, and this bound is related with the metric tensor $G^{M}$ and its relation with $G^{Mc}$. This bound is essentially the $C^{*}$-algebraic formulation of the Helstrom bound used in quantum information theory, and the content of the following proposition will be the key point to formulate the Helstrom bound in the $C^{*}$-algebraic framework.

**Proposition** **3.**
*Let (M,j,O) be a parametric model of states on the finite-dimensional $C^{*}$-algebra A, and let $G^{M}$ be the symmetric covariant tensor on M defined by Equation (Equation 21). Let $\left(M,j^{c},\Delta_{n}^{+}\right)$ be a parametric statistical model associated with (M,j,O), and let $G^{Mc}$ be the symmetric covariant tensor on M defined by Equation (Equation 62). Then, we have*
(121)$$G_{m}^{M}\left(v_{m},v_{m}\right)\;\geq \;G_{m}^{Mc}\left(v_{m},v_{m}\right)$$
*for every m∈M and every $v_{m}\in T_{m}M$.*


**Proof.** According to the definition of the SLD given in Equation (Equation 23), given an arbitrary tangent vector $v_{m}\in T_{m}M$, there is a gradient vector field $Y_{a}$ on O such that
(122)$$T_{m}j\left(v_{m}\right)\;=\;Y_{a}\left(\rho_{m}\right).$$Consequently, we have (recalling (Equation 16))
(123)$$G_{m}^{M}\left(v_{m},v_{m}\right)\;=\;G_{\rho_{m}}\left(Y_{a}\left(\rho_{m}\right),Y_{a}\left(\rho_{m}\right)\right)\;=\;\rho_{m}\left(a^{2}\right)-{\left(\rho_{m}\left(a\right)\right)}^{2}\;.$$On the other hand, by definition, we have
(124)$$G^{Mc}={\left(j^{c}\right)}^{*}G_{FR}\;=\;{\left(m^{*}\;\circ \;j\right)}^{*}G_{FR}\;=\;j^{*}\left({\left(m^{*}\right)}^{*}G_{FR}\right)\;,$$
which means
(125)$$G_{m}^{Mc}\left(v_{m},v_{m}\right)\;=\;{\left({\left(m^{*}\right)}^{*}G_{FR}\right)}_{\rho_{m}}\left(Y_{a}\left(\rho_{m}\right),Y_{a}\left(\rho_{m}\right)\right)\;,$$
and thus we have to prove that
(126)$${\left({\left(m^{*}\right)}^{*}G_{FR}\right)}_{\rho_{m}}\left(Y_{a}\left(\rho_{m}\right),Y_{a}\left(\rho_{m}\right)\right)\;\leq \;\rho_{m}\left(a^{2}\right)-{\left(\rho_{m}\left(a\right)\right)}^{2}$$
to prove the proposition.We note that, with any fixed ρ∈O and given an arbitrary non-zero gradient tangent vector $Y_{a}\left(\rho \right)$, there is an element $a_{c}\in C_{n}\equiv C\left(X_{n}\right)$ and a gradient tangent vector $Y_{a_{c}}\left(m^{*}\left(\rho \right)\right)$ at $m^{*}\left(\rho \right)\in O^{c}$ such that
(127)$$T_{\rho }m^{*}\left(Y_{a}\left(\rho \right)\right)\;=\;Y_{a_{c}}\left(m^{*}\left(\rho \right)\right),$$
and a direct computation shows that $a_{c}$ is characterized by the property
(128)$$\rho \left(\left\{a,m\left(b_{c}\right)\right\}\right)-\rho \left(a\right)\;\rho \left(m\left(b_{c}\right)\right)\;=\;\rho \left(m\left(a_{c}\;b_{c}\right)\right)-\rho \left(m\left(a_{c}\right)\right)\;\rho \left(m\left(b_{c}\right)\right)$$
for all $b_{c}\in C_{n}$. Therefore, we have
(129)$$\begin{array}{cc} {\left({\left(m^{*}\right)}^{*}G_{FR}\right)}_{\rho }\left(Y_{a}\left(\rho \right),Y_{a}\left(\rho \right)\right) & \;=\;{\left(G_{FR}\right)}_{m^{*}\left(\rho \right)}\left(T_{\rho }m^{*}\left(Y_{a}\left(\rho \right)\right),\;T_{\rho }m^{*}\left(Y_{a}\left(\rho \right)\right)\right)\;=\;\\ & \;=\;{\left(G_{FR}\right)}_{m^{*}\left(\rho \right)}\left(Y_{a_{c}}\left(m^{*}\left(\rho \right)\right),\;Y_{a_{c}}\left(m^{*}\left(\rho \right)\right)\right)\;=\;\\ & \;=\;\rho \left(m\left(a_{c}^{2}\right)\right)-{\left(\rho (m\left(a_{c}\right))\right)}^{2}\;. \end{array}$$Recalling Equation (Equation 126), we see that if the inequality
(130)$$\rho \left(m\left(a_{c}^{2}\right)\right)-{\left(\rho (m\left(a_{c}\right))\right)}^{2}\;\leq \;\rho \left(a^{2}\right)-{\left(\rho (a)\right)}^{2}$$
holds for all $\rho ,a,a_{c}$ and m satisfying Equation (Equation 127), then the proposition is proved.Next, by means of Equation (Equation 128), we write
(131)$$\rho \left(m\left(a_{c}^{2}\right)\right)-{\left(\rho (m\left(a_{c}\right))\right)}^{2}\;=\;\rho \left(\left\{a,m\left(a_{c}\right)\right\}\right)-\rho \left(a\right)\;\rho \left(m\left(a_{c}\right)\right)\;,$$
and since ρ({·,·})−ρ(·)ρ(·) is an inner product on the space of self-adjoint elements of A, we may apply the Cauchy–Schwarz inequality to obtain
(132)$${\left(\rho \left(m\left(a_{c}^{2}\right)\right)-{\left(\rho (m\left(a_{c}\right))\right)}^{2}\right)}^{2}\;\leq \;\left(\rho \left(a^{2}\right)-{\left(\rho \left(a\right)\right)}^{2}\right)\;\;\left(\rho \left(m\left(a_{c}\right)\;m\left(a_{c}\right)\right)-{\left(\rho (m\left(a_{c}\right))\right)}^{2}\right)\;.$$Now, m is a positive unital map, and thus it satisfies Kadison’s inequality
(133)$$m\left(a_{c}^{2}\right)\;\geq m\left(a_{c}\right)\;m\left(a_{c}\right),$$
from which it follows that
(134)$$\rho \left(m\left(a_{c}^{2}\right)\right)\;\geq \;\rho \left(m\left(a_{c}\right)\;m\left(a_{c}\right)\right)\;.$$Consequently, assuming that $\rho \left(m\left(a_{c}^{2}\right)\right)-{\left(\rho (m\left(a_{c}\right))\right)}^{2}\neq 0$, we have
(135)$$\frac{\rho \left(m\left(a_{c}\right)\;m\left(a_{c}\right)\right)-{\left(\rho (m\left(a_{c}\right))\right)}^{2}}{\rho \left(m\left(a_{c}^{2}\right)\right)-{\left(\rho (m\left(a_{c}\right))\right)}^{2}}\;\leq \;1$$
and thus
(136)$$\begin{array}{cc} \rho \left(m\left(a_{c}^{2}\right)\right)-{\left(\rho (m\left(a_{c}\right))\right)}^{2} & \;\leq \;\rho \left(a^{2}\right)-{\left(\rho \left(a\right)\right)}^{2}\;, \end{array}$$
and the proposition is proved. □

From the proof of Proposition 3, we easily obtain the following corollary.

**Corollary** **1.**
*Let (M,j,O) be a parametric model of states on the $C^{*}$-algebra A. Suppose there is a unital, Abelian $C^{*}$-subalgebra C⊆A such that, for all $v_{m}\in T_{m}M$, the SLD $Y_{a}\left(\rho_{m}\right)$ of $v_{m}$ at $\rho_{m}=j\left(m\right)$ given by*
(137)$$T_{m}j\left(v_{m}\right)\;=\;Y_{a}\left(\rho_{m}\right)$$
*is such that a∈C. Suppose also that the measurement procedure $m:=i_{C}$ given by the natural inclusion of C in A gives rise to a parametric statistical model $\left(M,j^{c},\Delta_{n}^{+}\right)$ associated with (M,j,O). Then, it holds*
(138)$$G_{m}^{M}\left(v_{m},v_{m}\right)\;=\;G_{m}^{Mc}\left(v_{m},v_{m}\right).$$


Now, let (M,j,O) be a parametric model of states on the $C^{*}$-algebra A, and let $\left(M,j^{c},\Delta_{n}^{+}\right)$ be a parametric statistical model associated with (M,j,O). Assume (M,j,O) and $\left(M,j^{c},\Delta_{n}^{+}\right)$ to be such that $G^{M}$ and $G^{Mc}$ are invertible. Let *C* be a cost function and E an estimator as in Section 5. Assume E is a stationary estimator at $m_{*}$, and let $C_{E_{j}}:M\to R$ be the smooth function given by $C_{E_{j}}\left(m\right):=C\left(m,E_{j}\right)$, where $E_{j}\equiv E\left(x_{j}\right)$ with $x_{j}\in X_{n}$.

According to the results of Section 6 (see Equations (Equation 98), (Equation 103), and (Equation 114)), given $v_{m_{*}},w_{m_{*}}\in T_{m_{*}}M$, the bilinear form
(139)$$C\left(v_{m_{*}},w_{m_{*}}\right)\;:=\;\sum_{j=1}^{n}\;v_{m_{*}}\left(C_{E_{j}}\right)\;w_{m_{*}}\left(C_{E_{j}}\right)\;p^{j}\left(m_{*}\right),$$
where $v_{m_{*}}\left(C_{E_{j}}\right)$ is the derivative of $C_{E_{j}}$ in the direction of $v_{m_{*}}$ evaluated at $m_{*}\in M$ (and similarly for $w_{m_{*}}\left(C_{E_{j}}\right)$), satisfies the Cramer–Rao bound given by
(140)$$C\left(v_{m_{*}},v_{m_{*}}\right)\;\geq \;{\left(G_{m_{*}}^{Mc}\right)}^{-1}\;\left(H_{m_{*}}\left(v_{m_{*}}\right),\;H_{m_{*}}\left(v_{m_{*}}\right)\right),$$
where $G^{Mc}$ is the Fisher–Rao metric on *M* seen as a parametric statistical model in $\Delta_{n}^{+}$, and $H_{m_{*}}$ is the Hessian form of the function $L_{m_{*}}:M\to R$ given by $L_{m_{*}}\left(m_{1}\right):=L\left(m_{1},m_{*}\right)$ at the point $m_{1}=m_{*}$ (see Equation (Equation 78)).

Then, Proposition 3 states that
(141)$$G_{m}^{M}\left(w_{m},w_{m}\right)\;\geq \;G_{m}^{Mc}\left(w_{m},w_{m}\right)$$
for every $w_{m}\in T_{m}M$. Consequently, we also obtain that
(142)$${\left(G_{m}^{M}\right)}^{-1}\left(\alpha_{m},\alpha_{m}\right)\;\leq \;{\left(G_{m}^{Mc}\right)}^{-1}\left(\alpha_{m},\alpha_{m}\right)$$
for every $\alpha_{m}\in T_{m}^{*}M$ (see ([105] Ex. 1.2.12)), and the Cramer–Rao bound in Equation (Equation 116) allows us to state that
(143)$$C\left(v_{m_{*}},v_{m_{*}}\right)\;\geq \;{\left(G_{m_{*}}^{Mc}\right)}^{-1}\;\left(H_{m_{*}}\left(v_{m_{*}}\right),\;H_{m_{*}}\left(v_{m_{*}}\right)\right)\;\geq \;{\left(G_{m_{*}}^{M}\right)}^{-1}\;\left(H_{m_{*}}\left(v_{m_{*}}\right),\;H_{m_{*}}\left(v_{m_{*}}\right)\right).$$We proved the following.

**Proposition** **4.**
*Let $\left(M,j^{c},\Delta_{n}^{+}\right)$ be the parametric statistical model associated with a parametric model of states (M,j,O). Assume that both $G^{M}$ and $G^{Mc}$ are invertible. Let C be a cost function and let E be a stationary estimator for C at m. If the Hessian form of $L_{m_{*}}$ at $m_{*}$ is invertible, then we have the generalized Helstrom bound*
(144)$$\mathrm{Cov}\left(\xi_{m_{*}},\xi_{m_{*}}\right)\;\geq \;{\left(G_{m_{*}}^{Mc}\right)}^{-1}\;\left(\xi_{m_{*}},\;\xi_{m_{*}}\right)\geq \;{\left(G_{m_{*}}^{M}\right)}^{-1}\;\left(\xi_{m_{*}},\;\xi_{m_{*}}\right)$$
*for all $\xi_{m_{*}}\in T_{m*}^{*}M$.*


This is the ***Helstrom bound*** for parametric models of states on a $C^{*}$-algebra. Indeed, when A is the algebra B(H) of bounded operators on the Hilbert space H of a finite-level quantum system, O is the orbit of faithful density operators on H, *M* is an open subset of some $R^{k}$ with k∈N. Then, in accordance with Remark 2, the cost function *C* may be taken to be the Euclidean distance on $R^{k}\times R^{k}$ pulled back on M×M, and a direct computation shows that Equation (Equation 143) reduces to the so-called ***Helstrom bound*** used in quantum estimation theory or quantum metrology [6,7,8,90].

**Remark** **4**(Helstrom bound for multiple-round models). *If we consider multiple rounds as in the end of Section 4, that is, we set $X=Y^{N}$, then Proposition 1 implies that the Helstrom bound can be written as*
(145)$$\begin{array}{cc} C(v_{m},v_{m}) & \geq {\left(G_{m}^{McN}\right)}^{-1}\left(H_{m}\left(v_{m}\right),H_{m}\left(v_{m}\right)\right)\\ & \geq {\left(G_{m}^{MN}\right)}^{-1}\left(H_{m}\left(v_{m}\right),H_{m}\left(v_{m}\right)\right)\\ & \geq \frac{1}{N}\;{\left(G_{m}^{M}\right)}^{-1}\;\left(H_{m}\left(v_{m}\right),H_{m}\left(v_{m}\right)\right), \end{array}$$*and this equation allows for the asymptotic analysis of the bound.*

The Helstrom bound is a universal bound for all the possible parametric statistical models associated with a given parametric model of states on a given $C^{*}$-algebra. This makes it quite a remarkable bound.

It is clear that, independently of the cost function and of the estimator we may choose, the Helstrom bound may be saturated if and only if
(146)$$G_{m}^{M}\left(v_{m},v_{m}\right)\;=\;G_{m}^{Mc}\left(v_{m},v_{m}\right).$$Then, Corollary 1 shows that this is in principle always true for one-dimensional models because we can always take the unital, Abelian $C^{*}$-subalgebra generated by the self-adjoint element a associated with the SLD of a given $v_{m}$ at $\rho_{m}$, and we are in the hypothesis of the corollary. However, it is also clear that for higher-dimensional models like the one in Example 4, this strategy may not be available.

## 8. Conclusions

We presented a preliminary account of the formulation of estimation theory in the context of parametric models of states on finite-dimensional $C^{*}$-algebras. The aim is to set the stage for the development of a mathematical formulation of estimation theory that can deal with the classical and quantum case “at the same time” by simply switching between commutative and noncommutative algebras.

After reviewing the differential geometric properties of the space of states S of an arbitrary finite-dimensional $C^{*}$-algebra A, we introduced the notion of parametric model of states on A. Then, following what is done in quantum information theory using POVMs, we considered how the explicit choice of a positive linear map from A to a suitable commutative $C^{*}$-algebra C gives rise to the notion of parametric statistical model of states associated with the starting parametric model of states on A. This parametric statistical model may be viewed as a classical-like snapshot of the given parametric model of states on the possibly noncommutative algebra A, and the Cramer–Rao bound for manifold-valued estimators is available for this model.

The fact that when A is noncommutative there is more than one such classical-like snapshot means that there is a Cramer–Rao bound for every classical-like snapshot of a given parametric model of states on A. This instance leads us to reformulate the so-called Helstrom bound to the case of a parametric model of states on a generic $C^{*}$-algebra and not just the algebra of bounded linear operators on a Hilbert space as it is customarily done in quantum information theory. The Helstrom bound gives a lower bound for all the possible Cramer–Rao bounds associated with the classical-like snapshots of a given parametric model of states on A. The possibility of considering also multiple-round models is briefly discussed, and the Helstrom bound derived in this context will be the starting point for the asymptotic theory of estimation theory in the $C^{*}$-algebraic framework we will deal with in future works.

As already remarked in the introduction, this work should be interpreted as a preliminary step toward a more general understanding of classical and quantum estimation theory. Accordingly, there are different instances that are left open for further developments. For instance, it is necessary to understand the general conditions for the attainability of the Helstrom bound for parametric models of states of dimension greater or equal than 2. It is also necessary to understand how to formulate other relevant bounds like the RLD-bound and the Holevo bound used in quantum information theory in the $C^{*}$-algebraic framework, as well as to understand how to perform the transition to the infinite dimensional case. From another point of view, it would be interesting to understand a suitable $C^{*}$-algebraic counterpart of the Amari–Cencov 3-tensor and the affine geometry it encodes in order generalize to the quantum case the understanding of the role of Frobenius manifolds recently investigated in the classical case [106,107]. We plan to address these issues in future works.

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
