# Peer review of "Differential Geometric Aspects of Parametric Estimation Theory for States on Finite-Dimensional C∗-Algebras"

_entropy, 2020, doi:10.3390/e22111332_

Round 1
Reviewer 1 Report
- For future research, the authors might like to exploit the work on finite dimensional nonlinear filters by Sanjoy Mitter, Roger Brockett, Steve Marcus, Daniel Ocone, Hector Sussmann, et al. about 40 years ago. In particular, they characterize the existence of finite dimensional nonlinear filters in terms of the dimension of the observation Lie algebra generated by the operators in the Zakai equation. The three simple prototypical examples are:(a) the Kalman filter (1960) which has the Heisenberg Lie algebra; (b) the Benes filter (1981) which is equivalent to the Kalman filter through a gauge transformation; and (c) Murray Wonham's filter (1965) for finite state Markov processes (this example is the closest to the paper under review). For example, one could generalize this work to the quantum case using the machinery in the paper under review.
- For future research, the authors might like to exploit the work of Persi Diaconis and Ulf Grennander on classicial probability models for various algebraic structures.
- The authors should explain why they do not need to assume nowhere vanishing densities or smoothness as hypotheses for their theorems on the Cramer-Rao bound; it appears that such assumptions are hidden by the overall assumption of linear functionals in the C* algebra. Otherwise, readers who are familiar with the bound in Cramer's book will be confused. Also, the hypothesis that the support of the likelihood does not depend on the measurements is hidden; the authors should explain why this is so. Without this assumption, the CRB is not correct, but rather it is replaced by another slightly more complex bound that includes extra terms from differentiating the boundary of the integral; this is called the Leibnitz-Cramer-Rao bound, owing to the use of the Leibnitz rule to derive it.
- There are a few spelling errors sprinkled throughout the text.
- This paper is very well written, and it contains new and significant ideas with many fine references; it will be welcome by many readers of ENTROPY.
Author Response
First of all, we would like to thank the referee for the appreciation of our work and for suggesting a very interesting direction for future research regarding the reformulation of finite-dimensional nonlinear filter theory in the context of C*-algebras. At the moment, we are not ready to comment on the subject, but we will definitely try to understand the extent to which it is possibile to adapt this classical theory in the quantum framework.
Regarding the work of Persi Diaconis and Ulf Grennander, we will follow the referee’s advice and investigate how their work may (hopefully positively) “affect” ours.
Regarding the third point in the report, we note that, according to definition 4 and remark 1 in
the manuscript, the very definition of the parametric statistical model we use in section 6 on the
Cramer-Rao bound already implies the fact that the probability distributions and their associated probability density functions we consider are nowhere-vanishing and smooth. Similar observations apply to the hypothesis that the support of the likelihood does not depend on the measurements.
As correctly pointed out by the referee, these regularity properties depend on the fact that we are looking at probability distributions as particular linear functional on a finite-dimensional commutative C*-algebra. In accordance with the referee, we realized that these considerations may not be as transparent as they should, and thus we added clarifying sentences in section 4 between definition 4 and remark 1, right after the first paragraph in section 6 (the one on the Cramer-Rao bound), and right after proposition 2 (the Cramer-Rao bound) in section 6.
Regarding the spelling errors sprinkled throughout the text mentioned by the referee, we did our best to correct all we were able to detect.
Reviewer 2 Report
A geometrical formulation of estimation theory for finite-dimensional C^*
algebras is presented. This formulation allows to deal with the classical and quantum case in a single, unifying mathematical framework. The derivation of the Cramer-Rao and Helstrom bounds for parametric statistical models with discrete and finite outcome spaces is presented.
I think that paper can be accepted as it is but I advitize to have a look on relation with affine algebraic geometry.
I recommend to mention A. Ya. Kanel-Belov, M. L. Kontsevich, “The Jacobian conjecture is stably equivalent to the Dixmier conjecture”, Mosc. Math. J., 7:2 (2007), 209–218 , and also review dedicated to memory of Yagzev :Alexei Belov, Leonid Bokut, Louis Rowen, Jie-Tai Yu, “The Jacobian Conjecture, Together with Specht and Burnside-Type Problems”, Automorphisms in Birational and Affine Geometry (Bellavista Relax Hotel, Levico Terme –Trento, October 29th – November 3rd, 2012, Italy), Springer Proceedings in Mathematics & Statistics, 79, Springer Verlag, 2014, 249–285
Author Response
We thank the referee for pointing out the references on the Jacobian conjecture. However, we were not able to mention said references because we are not able to clearly see the connection between the Jacobian conjecture, the affine algebraic geometry, and our work.